# Genetic analysis of seed traits in *Sorghum bicolor* that affect the human gut microbiome

Qinnan Yang [1,2], Mallory Van Haute[1,2], Nate Korth[2,3], Scott E. Sattler[4,5], John Toy[4,5], Devin J. Rose [1,2,5], James C. Schnable [2,5,6] & Andrew K. Benson [1,2] ✉

Prebiotic fibers, polyphenols and other molecular components of food crops significantly affect the composition and function of the human gut microbiome and human health. The abundance of these, frequently uncharacterized, microbiome-active components vary within individual crop species. Here, we employ high throughput in vitro fermentations of pre-digested grain using a human microbiome to identify segregating genetic loci in a food crop, sorghum, that alter the composition and function of human gut microbes. Evaluating grain produced by 294 sorghum recombinant inbreds identifies 10 loci in the sorghum genome associated with variation in the abundance of microbial taxa and/or microbial metabolites. Two loci co-localize with sorghum genes regulating the biosynthesis of condensed tannins. We validate that condensed tannins stimulate the growth of microbes associated with these two loci. Our work illustrates the potential for genetic analysis to systematically discover and characterize molecular components of food crops that influence the human gut microbiome.

Over the last six decades, the incidence of complex lifestyle diseases such as obesity, diabetes, metabolic disease, and inflammatory bowel diseases have grown at alarming rates in countries with westernized diets[1–4]. In addition to genetic and environmental factors, many diseases in these categories are also associated with disconfiguration of the human gastrointestinal microbiome[5]. In instances such as obesity and metabolic liver diseases, disconfigured microbiomes have been shown to be causal to disease processes[6,7]. Dietary factors are also associated with these same diseases and diet has a major effect on taxonomic configuration and function of the gut microbiome[8,9]. Consequently, there is tremendous interest in developing novel foods and novel food ingredients that could be used to manipulate the gut microbiome in predictable ways to reduce susceptibility to diseases[10].

Several types of bioactive molecules in human diets are known to affect taxonomic configuration and function of the gut microbiome, including complex carbohydrates and fibers, polyphenols, lipids, and seed proteins[11,12], but these components have not been systematically cataloged and there is a major gap in our understanding of how plant breeding and genetics can impact abundances of many of these bioactive molecules. This gap is significant because the goals of crop breeding and improvement programs have historically been focused on agronomic and yield traits, and more recently on sustainability traits such as carbon footprints and water use[13]. There is little understanding of whether emphasis on agronomic, yield, or sustainability traits may have trade-off effects on nutritional traits or traits that affect the gut microbiome. This gap and the mounting opportunity to

[1]Department of Food Science and Technology, University of Nebraska, Lincoln, NE, USA. [2]Nebraska Food for Health Center, University of Nebraska, Lincoln, NE, USA. [3]Complex Biosystems Graduate Program, University of Nebraska, Lincoln, NE, USA. [4]Wheat, Sorghum and Forage Research Unit, USDA-ARS, Lincoln, NE, USA. [5]Department of Agronomy and Horticulture, University of Nebraska, Lincoln, NE, USA. [6]Center for Plant Science Innovation, University of Nebraska, Lincoln, NE, USA. ✉e-mail: abenson1@unl.edu

improve nutritional and health-promoting traits is becoming more broadly recognized within the plant science community[14].

Sorghum (*Sorghum bicolor* (L.) Moench) is the fifth most widely produced grain globally, behind maize, rice, wheat, and barley, and it is a staple in the diets of populations in some semi-arid areas (estimated 500 million people) but is also widely grown in the United States where it is valued for its ability to produce grain in areas with insufficient water for maize production. Sorghum populations exhibit substantial genetic and phenotypic diversity, including diversity in the abundance of bioactive components[15,16]. For example, substantial variation between sorghum varieties exists for several phenotypes such as grain color (polyphenols) and fiber content that may have effects on the human gut microbiome[17–20]. Thus, sorghum serves as an excellent model system to begin study of seed traits that affect human gut microbiome fermentation.

In this work, we develop an approach for genetic analysis of seed traits that affect fermentation patterns by the human gut microbiome in a model food crop (sorghum) to address the gap and the opportunity to improve nutritional traits. We show how these analyses can be used to systematically study variation in microbiome-active components within any food crop species. Our study highlights how existing genetic resource populations of food crop species can be exploited for co-analysis of seed traits and microbiome traits to efficiently pinpoint candidate loci and pathways through which genetic variation can affect the human gut microbiome. Ultimately, this approach will pave the way to incorporate microbiome traits into crop improvement programs to improve human health.

## Results

### Genetic analysis of seed traits that affect the human gut microbiome

An overview of our approach for genetic analysis of seed traits that can influence human gut microbiome is illustrated in Fig. 1. We focused this proof-of-concept study on a well-characterized set of sorghum recombinant inbred lines (RILs, Fig. 1a) derived from two genetically diverse parents (IS3620C from the Guinea sorghum subpopulation and BTx623, which is a blend of the Kafir and Caudatum sorghum subpopulations)[21,22]. Grain from individual RILs was used for automated in vitro microbiome screening (AiMS) to quantify interaction of human gut microbes with pre-digested grain from the RILs (see "Methods" for details). The AiMS method first prepares the grain samples from each RIL through a series of steps that resemble the digestive process (Fig. 1b), including milling (mastication), hydrolysis by acid and digestive enzymes (digestion), and dialysis (absorption in the small intestine). The remaining components of the digested grain are then mixed with aliquots of a stool microbiome (Fig. 1c) from an individual human donor and incubated anaerobically (Fig. 1d) to mimic interaction of the digested grain components with the colonic microbiome. Abundances of microbial taxa in the individual fermentations are measured by 16S rDNA sequencing and used as quantitative traits for quantitative trait locus (QTL) mapping analysis (Fig. 1e), where genetic variation at specific regions in the sorghum genome in the RILs is tested for association with significant differences in abundances of microbial taxa in the fermentations.

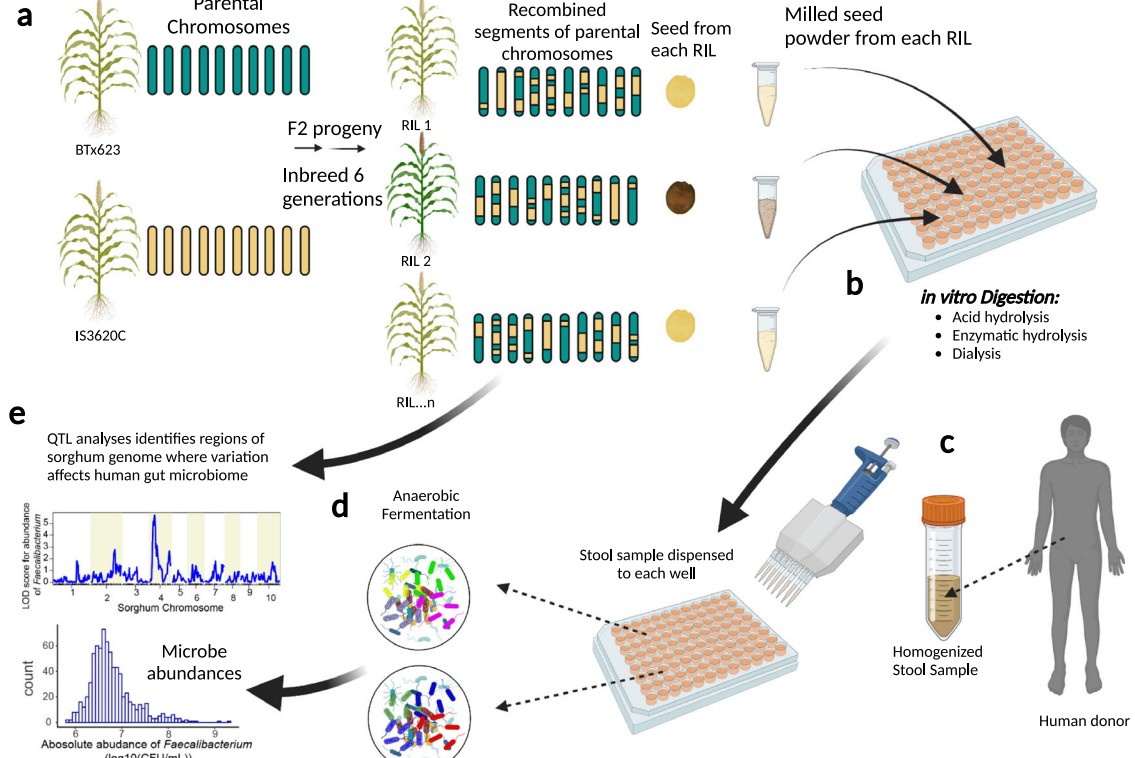

**Fig. 1 | Schematic diagram of the of AiMS for genetic analysis of microbiome traits. a** This section of the diagram depicts the well-characterized set of sorghum recombinant inbred lines derived from two genetically diverse parents and illustrates how each individual derived recombinant inbred line (RIL) carries random combinations of genomic segments from both parents. **b** This section of the diagram depicts how the AiMS method first prepares the grain from each RIL through a series of steps that resemble the digestive process—milling (mastication), acid and enzymatic hydrolysis (digestion), and dialysis (absorption in the small intestine). **c, d** The remining components of the digested grain from each RIL are then mixed with aliquots of a stool microbiome from an individual human donor and incubated anaerobically to mimic interaction of the digested grain components with the colonic microbiome. **e** Abundances of microbial taxa in the individual fermentations are then estimated by 16S rDNA sequencing and used as quantitative traits for genetic analysis by Quantitative Trait Locus (QTL) mapping to identify regions in the sorghum genome where variation has significant effects on abundances of taxa in the fermentations. The diagram was created with BioRender.com.

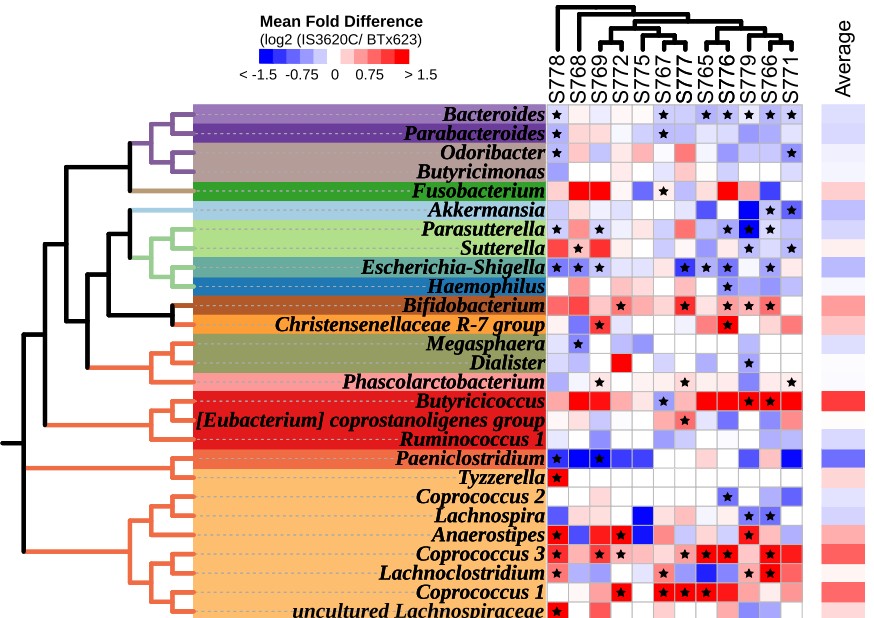

**Fig. 2 | Shifts in the abundances of bacterial taxa in AiMS reactions with grain from the IS3620C or BTx623 parental lines.** IS3620C and BTx623 are the parental lines of sorghum used to generate the RIL population used in our study. The left-hand portion of the figure illustrates microbial taxa from 12 different human subjects that show significant differences in abundances between fermentation of grain from the IS3620C or BTx623 parents. The microbial taxa are organized based on a neighbor-joining tree representing their phylogenetic relationships. These phylogenetic relationships were inferred from representative sequences of species from each bacterial genus using MUltiple Sequence Comparison by Log- Expectation (MUSCLE). Bacterial genera from the same family are shaded in the same colors. On the right-hand portion of the figure, abundance differences of each taxon in AiMS fermentations from each subject are depicted by heat-mapping of the mean log2-transformed-fold difference from fermentation of grain from the IS3620C parent relative to BTx623 parent. All taxa illustrated show significant microbiome-wide effects across the human subjects (unadjusted $p < 0.05$; two-sided rANOVA) and statistically significant differences between IS3620C and BTx623 sorghum for each taxon were determined by subject microbiome (Two-sided Wilcoxon test; $p < 0.05$) and denoted by an asterisk. Source data are provided as a Source Data file.

## Parental lines of the RILS have distinct effects on human gut microbiomes

To define differences in seed characteristics between the two parental lines that affect the human gut microbiome fermentation patterns, we first evaluated AiMS fermentations of grain from the BTx623 and IS3620C parental lines across microbiomes from 12 different healthy human donors (see Methods for human subjects). Baseline (before fermentation) composition of the microbiomes from each of the 12 human donors are shown in Supplementary Fig. 1. Permutational multivariate analysis of variance (PERMANOVA) based on β-diversity using Bray-Curtis distances of the microbiome after fermentation showed that 10 of the 12 microbiomes had significant differences in overall microbiome composition after fermentation ($p < 0.05$), with parental line explaining 27–53% of the observed variation (Supplementary Fig. 2a), suggesting differences in grain composition between the BTx623 and IS3620C parental lines drive distinct fermentation patterns of the gut microbiome.

The individual taxonomic responses of the ten microbiomes to the parental lines were donor microbiome-specific, and collectively 27 microbial clades showed significant parental line-specific differences in abundances ($p < 0.05$, unadjusted repeated measures analysis of variance (ANOVA), Fig. 2). Nine clades showed consistent, differential responses to the parental grain lines in microbiomes of multiple (three or more) donors ($p < 0.05$, Wilcoxon test). For example, the clades *Coprococcus 1*, *Coprococcus 3* and *Bifidobacterium* were more abundant in fermentations of IS3620 grain than BTx623 grain across microbiomes from 4, 7 and 5 donors, respectively (Fig. 2). In contrast, *Escherichia*, *Parasutterella* and *Bacteroides* were more abundant across 7, 5 and 7 donor microbiomes in fermentation of BTx623 than in IS3620C, respectively (Fig. 2).

## QTLs in the sorghum genome that affect human gut microbes

The microbiome of donor subject S765 showed significant overall difference (Supplementary Fig. 2a) between parental lines and many of the individual taxonomic responses observed in this microbiome were consistent with the population-wide average responses observed across all 12 donors (Supplementary Fig. 2b). This microbiome was subsequently used for AiMS-based phenotyping of 294 individual RILs from the BTx623 x IS3620C RIL population.

Across the microbial profiles of AiMS reactions from all 294 RILs, a total of 84 genera were represented by at least 100 reads in the resulting 16S microbiome data after rarefaction to 15,452 reads per sample and these taxa were used for subsequent genetic analyses. After log 10 transformation, observed abundances for many of these taxa across fermentation reactions of the 294 RILs were not-normally distributed (Supplementary Data 1; Shapiro-Wilk). Consequently, we used a non-parametric model for QTL mapping with the log trans-formed abundances of individual taxa (Supplementary Data 2) serving as phenotypes and genotypes from a previously published set of genetic markers scored across the BTx623 x IS3620C RIL population[22]. A total of 26 significant QTLs ($p < 0.05$, 1000 permutations) and several suggestive QTLs ($p < 0.1$, 1000 permutations) were identified that influence the abundances of 19 different microbial clades (Supplementary Table 1). These QTLs were positioned on nine of the ten sorghum chromosomes, illustrating a polygenic architecture of genetic variation in sorghum can influence human microbiome phenotypes.

Genetic associations were detected for taxa from three phyla (*Bacteriodetes*, *Firmicutes*, and *Proteobacteria*) with 23 of the 26 significant QTLs corresponding to members of the *Firmicutes* phylum. Among these 23 significant QTLs, 14 correspond to genera in the *Ruminococcaceae* or *Lachnospiraceae* families. Genera from these

**Table 1 | Major effect loci for microbial taxa and short-chain fatty acids**

| Microbial taxa and short-chain fatty acid | LOD score | Chromosome | Peak position (Mb) | Confidence interval (Mb) |
|---|---|---|---|---|
| *Ruminococcaceae_Ruminococcaceae UCG.002* | 3.43 | 2 | 7.93 | 7.01–12.79 |
| *Peptostreptococcaceae_Paeniclostridium* | 3.22 | 2 | 9.69 | 4.83–77.22 |
| *Christensenellaceae_ChristensenellaceaeR.7group* | 4.17 | 2 | 9.69 | 7.93–9.69 |
| *Prevotellaceae_Paraprevotella* | 4.05 | 2 | 65.54 | 63.91–69.84 |
| Butyrate | 4.84 | 2 | 65.69 | 65.54–67.24 |
| Valerate | 3.73 | 2 | 65.69 | 65.54–67.24 |
| *Lachnospiraceae_Dorea* | 3.61 | 3 | 4.04 | 3.81–7.46 |
| *Lachnospiraceae_Coprococcus 3* | 3.88 | 3 | 4.04 | 3.81–72.46 |
| *Lachnospiraceae_Roseburia* | 3.52 | 3 | 71.51 | 4.04–73.24 |
| *Christensenellaceae_ChristensenellaceaeR.7group* | 3.63 | 3 | 71.51 | 71.51–72.69 |
| *Peptostreptococcaceae_Paeniclostridium* | 8.64 | 4 | 59.46 | 58.94–60.16 |
| *Prevotellaceae_Paraprevotella* | 4.01 | 4 | 61.16 | 60.49–61.88 |
| *Christensenellaceae_ChristensenellaceaeR.7group* | 7.37 | 4 | 61.3 | 60.84–61.88 |
| *Ruminococcaceae_Ruminococcaceae UCG.002* | 4.74 | 4 | 61.56 | 60.84–62.37 |
| *Lachnospiraceae_Coprococcus1* | 4.09 | 4 | 61.88 | 61.16–62.4 |
| *Lachnospiraceae_Roseburia* | 5.41 | 4 | 61.88 | 48.57–62.4 |
| *Ruminococcaceae_Faecalibacterium* | 7.02 | 4 | 61.88 | 61.16–62.4 |
| *Erysipelotrichaceae_Catenibacterium* | 5.26 | 4 | 61.88 | 59.46–62.37 |
| Valerate | 3.51 | 5 | 13 | 7.45–49.41 |
| *Lachnospiraceae_Coprococcus 3* | 3.04 | 5 | 47.69 | 7.45–54.78 |
| *Lachnospiraceae_Blautia* | 3.07 | 5 | 47.69 | 13–55.7 |
| *Lachnospiraceae_Dorea* | 3.19 | 5 | 47.69 | 13–55.7 |
| Butyrate | 3.08 | 5 | 47.69 | 13–55.7 |
| Propionate | 3.35 | 5 | 54.78 | 52.23–55.7 |

families, particularly *Faecalibacterium* and *Roseburia*, are increasingly being recognized as beneficial organisms in the microbiome that reduce susceptibility to inflammatory diseases[23–26]. Importantly, while *Faecalibacterium* and *Roseburia* showed highly significant genetic associations with variation in the RIL population, they did not exhibit significant differences between the parental lines, suggesting these microbial phenotypes and genetic associations in the RILs could be due to transgressive segregation.

**Multiple effect loci are defined by overlapping QTLs for multiple microbial taxa and their metabolic products**

Among the 26 significant QTLs, overlapping QTLs for two or more microbial taxa were present on chromosome 2, chromosome 3, chromosome 4 and chromosome 5 (Table 1 and Fig. 3a), suggesting these QTL regions behave as multiple effect loci (MEL) where genetic variation at the QTL peaks in the sorghum RILs has pleiotropic effects on multiple microbial taxa. The peaks of genetic associations at the MEL on chromosome 4, corresponding to a genomic interval from 59–62 Mb, influenced the greatest number of microbial taxa, affecting the relative abundances of a diverse group of eight different microbial taxa ($p < 0.05$; 1000 permutations). Within this 59–62 Mb interval, the physical peaks for *Catenibacterium*, *Roseburia*, *Coprococcus 1*, and *Faecalibacterium* (61,878,324), were immediately adjacent to peaks for *Ruminococcaceae UCG.002* (61,555,802), *Christensenellaceae_R7 group* (61,304,986), *Paraprevotella* (61,161,519), and *Paeniclostridium* (59,462,696). These QTL peaks also exhibited some of the highest logarithm of the odds (LOD) scores, (LOD 8.64 for *Paeniclostridium*, LOD 7.37 for *Christensenellaceae_R7 group*, LOD 7.02 for *Faecalibacterium*, LOD 5.41 for *Roseburia*, LOD 5.26 for *Catenibacterium*, and LOD 4.73 for *Ruminococcaceae UCG 002*).

Two MELs were identified on chromosome 2 with a peak in the 7.9–9.6 Mb region that affected three genera (*Christensenellaceae_R7 group*, *Ruminococcaceae_UCG 002 group*, and *Paeniclostridium*) and

another peak in the 65.54–65.69658 Mb region affected *Paraprevotella*. Two MELs were identified on chromosome 3, one MEL with a peak at 4.04 Mb affecting two members of the *Lachnospiraceae* (*Dorea* and *Coprococcus 3*) and a second MEL with a peak at 71.5 Mb affecting another member of the family *Lachnospiraceae* (*Roseburia*) and a member of the *Christensenellaceae_R7 group*. A fifth MEL on chromosome 5 with a peak at 47.69 Mb affected three genera from the family *Lachnospiraceae* (*Coprococcus 3*, *Blautia*, and *Dorea*).

In addition to QTL mapping of microbial taxa, we also used metabolic end products of microbial fermentation as quantitative traits for QTL analysis, specifically focusing on concentrations of major short-chain fatty acid (SCFA) from the individual AiMS fermentations as functional phenotypes of microbiome. Significant QTLs were detected for propionate, butyrate, and valerate production (Table 1 and Fig. 3a). The overlapping QTL peaks on chromosome 2 at 65,688,971 bp for butyrate and valerate production both shared confidence intervals from 65.54–67.24 Mb, which overlapped with the QTL for *Paraprevotella*. QTLs on chromosome 5 within the confidence interval of 7.45–55.7 Mb were associated with propionate, butyrate, and valerate and overlapped with QTLs for abundances of *Coprococcus 3*, *Blautia*, and *Dorea*. Overlapping of QTLs for microbial taxa and SCFAs implies variation at these loci may drive significant effects on taxonomic abundances and metabolites from their fermentation activities. Interestingly, the most significant MEL on chromosome 4 (based on number of microbial taxa affected) did not show significant effects on SCFA, suggesting that the broad effect of this MEL on microbial taxa may confound the SCFA phenotypes or that the QTL may manifest in ways that do not significantly influence metabolic end products of fermentation.

**QTLs for seed color and tannin content overlap with MELs for microbiome phenotypes**

While both parental sorghum lines produce cream-colored seeds, many of the 294 RILs produce brown-colored seeds (Supplementary

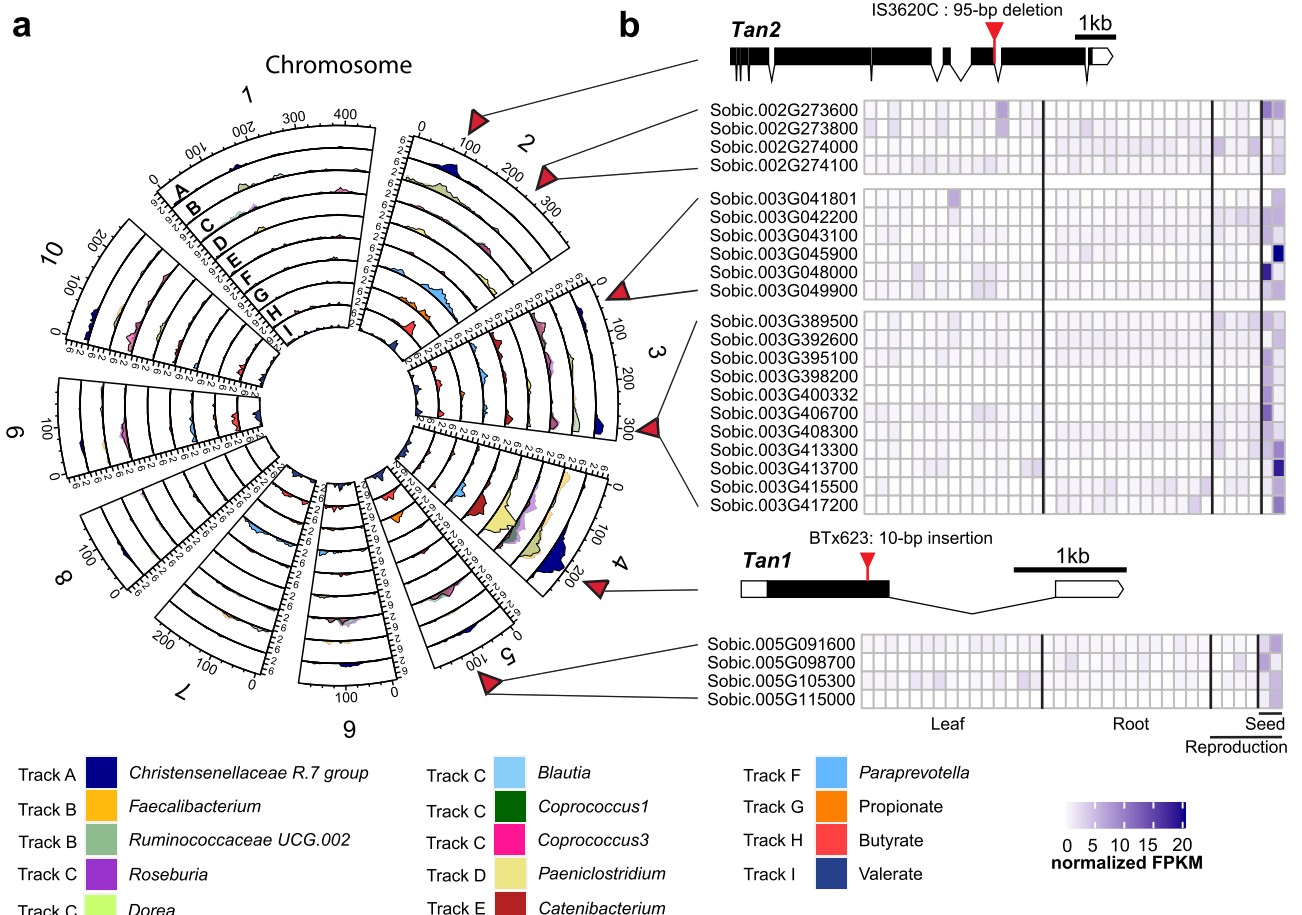

**Fig. 3 | QTL analysis of the RIL population using the S765 microbiome.**
**a** Logarithm of the odds (LOD) score profiles for microbial taxa showing significant QTLs in the RIL population. The circular plots illustrate each chromosome and positions are marked around the outer track in Centimorgans (cM). The individual tracks (A–I) depict the LOD plots of individual genera belonging to the same microbial family or SCFA that are color coded with the corresponding key shown at the bottom of the figure. LOD values are indicated on the *Y*-axis by chromosome for each LOD plot. The MELs on Chr 2, Chr 3, Chr 4, and Chr 5 are marked by red triangles on the outside of the diagram corresponding to the QTL peaks. **b** Heat map of tissue-specific gene expression data (from the Phytozome v13 database) for sorghum genes within the MELs on Chr 2, Chr 3, Chr 4, and Chr 5 is shown for candidate genes that are highly expressed in seed (normalized Fragments Per Kilobase of transcript per Million mapped reads (FPKM) > 5) and which have sequence variation between the two parental lines classified as having moderate or high impact on gene function. Gene structure of *Tan1* and *Tan2* and the causal loss of function mutations in two parental lines were also shown on the right. Source data are provided as a Source Data file.

Data 3), indicating seed color phenotypes segregate transgressively in the RIL population. QTL analysis of seed color using the first principal component of red, green and blue color identified major QTLs for seed color with major peaks on chromosome 2 (7.93 Mb) and chromosome 4 (61.5 Mb), but not other loci significantly associated with seed color, such as Y (chromosome 1), R (chromosome 3) and Z (chromosome 2) (Fig. 4a). The chromosome 4 QTL peak for seed color (61.5 Mb) overlaps with the physical location (61.1–61.8 Mb) of the MEL on chromosome 4 for eight microbial taxa and the QTL peak on chromosome 2 for seed color overlaps with the QTL peaks for the MEL on chromosome 2 at 4.83–12.79 Mb for three microbial taxa (Fig. 4a). The overlapping QTL peaks for seed color and traits that affect fermentation patterns by the human gut microbiomes implied that allelic variation at the 61.5 Mb region of chromosome 4 and 4.83–12.79 Mb region of chromosome 2 affects molecular components that contribute to seed color, and variation in these molecules may be driving the observed differences in the human gut microbiome.

The overlapping QTL peaks for seed color and for microbial taxa are very close to the physical locations of the *S. bicolor Tan2* (chromosome 2, 7.97 Mb) and *Tan1* (chromosome 4, 62.3 Mb) genes, which encode transcription factors that regulate the expression of genes in the polyphenol/flavonoid pathways, including proanthocyanidin and

anthocyanins that affect seed color in sorghum[27–29]. Grain color in sorghum is determined by the pigmentation of three distinct tissues. In the pericarp, or seed coat, two loci (Y and R, located on chromosomes 1 and 3 respectively) can condition white, yellow, or red seed color. Condensed tannins, conditioned by *Tan1* and *Tan2*, typically accumulate in the testa layer, below the pericarp, and are visible as a brown layer when the pericarp is thin or colorless as well as when seeds are ground into flour. Finally, the endosperm itself, the majority of the seed, can also vary in color between white and yellow. While the Y locus on chromosome 1 and R locus on chromosome 3 typically control seed color, proanthocyanidin (condensed tannins) can produce brown seed color when dominant alleles at both *Tan1* and *Tan2* occur in genetic backgrounds such as the spreader (S), which causes the spread of condensed tannins from the testa layer into the pericarp[20,27,29]. The *Tan1* gene (Sobic004G280800) is physically located at 62,315,396–62,318,779 and encodes a WD40-like protein while the *Tan2* gene on chromosome 2 (Sobic002G076600) is physically located at 7,975,937–7,985,221 and encodes an b-HLH-like transcription factor[28,30].

To confirm that seed color and microbiome traits associated with chromosome 2 and chromosome 4 are due to segregation at these loci in the RILs, we quantified tannin production from each of the 294 RIL

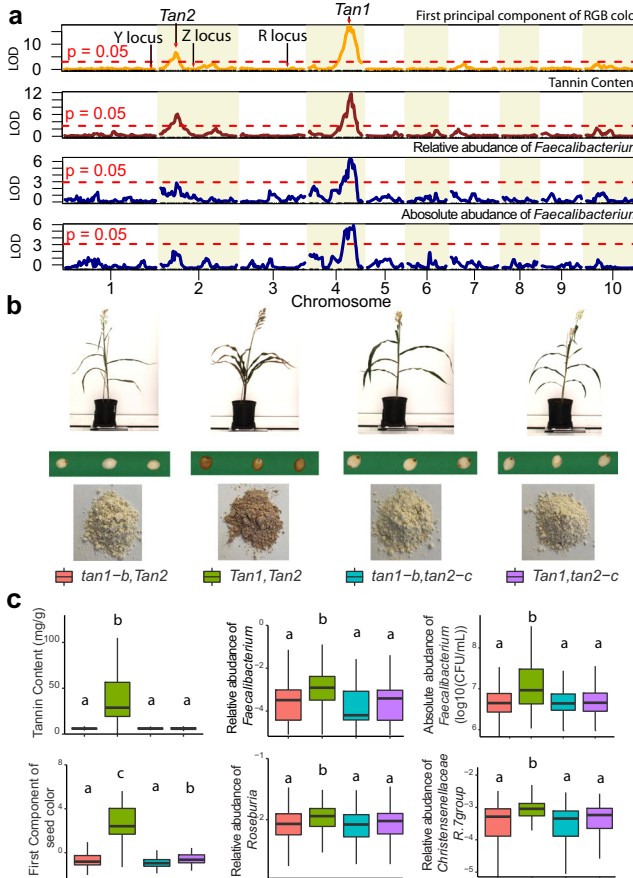

**Fig. 4 | *Tan1* and *Tan2* determine the seed color, tannin, and microbiome phenotypes in the RILs. a** Logarithm of the odds (LOD) score profiles for the first principal component of red, green, and blue color value (RGB), tannin content and the abundance of *Facecalibacterium* determined by 16S rRNA gene sequencing and by qPCR in the RIL population. The significance level was established by permutation testing (1,000 permutations). The locations of the R, Y, and Z loci known to affect pericarp color are marked with arrows on the LOD score profiles for the first principal component of RGB. **b** Each RIL was assigned to one of four genotypic categories based on haplotypes of markers closely linked to the *Tan1* and *Tan2* loci. The four genotypic categories correspond to tannin-negative categories (*tan1-b/ Tan2*; *Tan1/tan-2-c*; tan-1-b/tan-2-c) and the tannin-positive category (*Tan1/Tan2*). Pictures of whole plant, whole grain and color of powders from seed from a representative RIL in each category are illustrated. **c** Box and whisker plots are shown for phenotypic values of tannin concentration, seed color, relative abundance of individual microbial taxa (16S-amplicon-based abundances of *Faecalibacterium*, *Roseburia*, *Christensenellaceae R7 group*) and absolute abundance of *Faecalibacterium prausnitzii* by qPCR corresponding to RILs in each of the genotypic categories (*tan1-b/Tan2*, $n = 89$; *Tan1/Tan2*, $n = 45$; tan-1-b/tan-2-c, $n = 75$; *Tan1/tan-2-c*, $n = 45$). Displayed are the interquartile range (IQR; boxes), median (line), and 1.5 IQR (whiskers). Different letters (**a**–**c**) indicate significant difference among different genotypic categories determined by pairwise two-sided Kruskal–Wallis test corrected by Dunn's multiple comparisons ($p < 0.05$). Source data are provided as a Source Data file.

lines and from two parental lines (Supplementary Data 4). Like the trait values for seed color and microbiome traits, tannin production was also transgressive with 57 lines producing significant levels of tannins (10.34–118.08 mg/g seed) whereas the remaining 237 lines only produced low levels (0.09–9.60 mg/g) found in the parental lines (BTx623: 1.47 mg/g and IS3620C: 1.12 mg/g). Strong genetic associations with variation in tannin content among the sorghum lines used in this study were identified exclusively on chromosome 2 and chromosome 4 in regions overlapping with the peaks for seed color and specific microbial taxa (Fig. 4a). We did not detect any significant associations for seed

color any other loci, including the Y, R or Z, located on chromosome 1, 3 and 2, respectively (Fig. 4a), suggesting that the brown seed color in the RIL population used in this study was controlled solely by segregation of *Tan1* and *Tan2* alleles. Lastly, we also confirmed the overlap of microbiome, seed color, tannin content and absolute levels of *Faecalibacterium prausnitzii* using species-specific qPCR to more accurately quantify this organism in each of the AiMS reactions. As with the 16S rDNA data, QTL mapping of the absolute levels of this organism from qPCR data also gave significant associations with chromosome 4 and suggestive associations with chromosome 2, and these QTL peaks overlap with peaks from 16S rDNA data, seed color data, and tannin content (Fig. 4a). The overlap suggests that tannin content of the seed can drive differences in abundance of *F. prausnitzii* and other organisms influenced by these MELs.

**Segregation of dominant alleles at *Tan1* and *Tan2* loci explains transgressive segregation of seed color, tannin production, and microbiome phenotypes**

Functional products from both the *Tan1* and *Tan2* regulatory genes are required for tannin synthesis and high-impact mutations in either gene can mask effects of a dominant allele at the other locus (duplicate recessive epistasis), blocking tannin synthesis and yielding light-colored seed[28]. Neither BTx623 nor IS3620C produce significant quantities of tannins as a result of recessive loss of function alleles in the *Tan1* (BTx623) or the *Tan2* (IS3620C) genes. More specifically, BTx623 carries dominant (wild type) alleles of *Tan2*, but at the *Tan1* locus is homozygous for the *tan1-b* allele, which has a 10-base insertion in the C-terminal exon that truncates 35 amino acids from the C-terminus[30]. In contrast, the IS3620C parent carries dominant alleles at *Tan1*, but is homozygous for the *tan2-c* allele with a 95-base deletion that removes the entire intron between exons 7 and 8[28]. Thus, the homozygous inheritance of either or both of *tan1-b* (BTx623 parent) and *tan2-c* (IS3620C parent) alleles in the RILs would yield non-tannin phenotypes with light-colored seeds. In contrast, RILs inheriting wild-type *Tan1* (IS3620C parent) and *Tan2* (BTx623 parent) alleles would be expected to result in a tannin-positive phenotype, with brown seed color being observed in backgrounds that allow spread of the condensed tannins into the pericarp layers (transgressive phenotypes). If tannin production and function of *Tan1* and *Tan2* are indeed drivers of the QTLs for MELs of microbial taxa associated with the chromosome 2 and chromosome 4, then we would expect the transgressive phenotypes of the microbes to co-segregate with tannin production phenotype, brown seed color, and inheritance of dominant parental haplotypes linked to *Tan1* (IS3620C parent) and *Tan2* (BTx623).

When the RILs are grouped based on parental haplotypes of markers linked to *Tan1* and *Tan2*, we were able to predict parental haplotypes at *Tan1* and *Tan2* in 254 of the lines. Tannin production and dark seeds were exclusively found among the 45 RILs with haplotypes linked to wild-type *Tan1* alleles from the IS3620C parent and wild-type *Tan2* alleles from BTx623 (Fig. 4b, c and Supplementary Data 5) but not the other three haplotype combinations (*Tan1/tan2-c*, *tan1-b/Tan2*, and *tan1-b/tan2-c*). Like the seed color and tannin phenotypes, analysis of microbiome phenotypes across RILs in the four different genotypic classes at *Tan1* and *Tan2* showed that microbial genera associated with the chromosome 2 and chromosome 4 QTL peaks displayed the expected distribution across the four genotypic classes of RILs. *Christensenellaceae R7 group*, *Catenibacterium*, *Coprococcus 1*, *Roseburia*, *Paeniclostridium*, *Faecalibacterium*, and *Ruminococcaceae_UCG.002 group* all showed significantly higher abundances in tannin-producing RILs carrying wild-type haplotypes at the chromosome 4 *Tan1* (IS3620C) and chromosome 2 *Tan2* (BTx623) regions versus the other three genotypic categories carrying either *tan1-b* from BTx623, *tan2-c* from IS3620C, or both *tan1-b/tan2-c* from each parent

(Fig. 4c and Supplementary Data 5) and there were no significant differences between the tannin-negative genotypic categories. Thus, the microbiome phenotypes, seed color, and tannin production show the expected pattern of co-segregation with allelic variation at the *Tan1* and *Tan2* loci, and the gain of function expression of tannins in *Tan1* and *Tan2* RILs can explain the transgressive segregation of seed color, tannin, and microbial phenotypes.

## RILs and NILs with the same tannin phenotypes have similar effects on microbiomes from multiple human subjects

Since a representative microbiome of a single subject was used for QTL mapping, we next examined the ability to detect sorghum genetic variation (chromosome 2 and chromosome 4 MELs) driving microbial taxa against gut microbiomes from multiple human subjects. In these experiments, we pooled seeds from RILs based on haplotype of markers linked to *Tan1* and *Tan2* loci and tested them in AiMS reactions with microbiomes from the 11 other subjects used in the pilot study with parental lines. Thirty genera were detected with significant different abundance between *Tan1/Tan2* haplotype against other haplotypes (rANOVA analysis followed by false discovery rate (FDR) correction using Benjamini–Hochberg procedure, Fig. 5a). For example, *Tan1/Tan2* lines had significantly higher abundances of *Faecalibacterium*, *Christensenellaceae R7 group*, *Roseburia* and *Coprococcus 1* than lines carrying *tan1-b*, *tan2-c* or both alleles. On the other hand, the abundances of *Odoribacter, Paraprevotella, and Pasutterella* were significantly lower in *Tan1 Tan2* lines (Fig. 5a). Thus, the effects of variation at *Tan1* and *Tan2* appear to manifest similar taxonomic changes across microbiomes from different human subjects.

We further studied causal effects of variation at the tannin regulatory loci on gut microbes using grain from three pairs of near-isogenic lines (NILs). Each pair of NILS effectively differ only in their allelic content at the *Tan1* locus but are fixed for the wild-type allele at the *Tan2* locus. Seed from the pairs of NILs was used as substrate in AiMS reactions with microbiomes from the same 12 subjects. As illustrated in Supplementary Fig. 3, allelic variation at *Tan1* had significant effects on β-diversity of microbiome from subject 765 as well as the microbiomes from each of the other 11 donors. Several taxa across each of the 12 microbiomes showed significant responses to the tannin-positive against tannin-negative RILs and NILs (Fig. 5a, b). For example, the NILs homozygous for wild-type alleles at *Tan1* and RILs homozygous for wild-type haplotypes at *Tan1* and *Tan2* had significantly higher abundances of *Faecalibacterium*, *Christensenellaceae R7 group*, and *Roseburia* with consistent responses of these taxa in magnitude and directionality across the different microbiomes. Similarly, multiple microbiomes also showed decreased abundances of *Odoribacter, Paraprevotella*, and *Parasutterella* in tannin-positive RILs (Fig. 5a). Correlation analysis of genera with significantly different abundance between *Tan1/Tan2* haplotype and all other haplotypes in AiMS reactions across microbiomes from all 12 subjects also showed significant correlation of multiple taxa from 14 different families that aggregate to 5 different phyla (Fig. 5c, Pearson correlation coefficient of 0.65, $p < 4.3 - 05$). Thus, the *Tan1* alleles in the NILs can drive similar microbiome responses associated with the *Tan1* alleles in the RILs across the unique microbiome context of multiple subjects.

## Introduction of purified condensed tannins into fermentations with tannin-negative RILs restores the microbiome phenotype

Because the *Tan1* and *Tan2* loci regulate the pathways for anthocyanin and proanthocyanidin synthesis as well as other pathways, we used molecular complementation experiments with extracted condensed tannins to determine if the *Tan1/tan1-b*-associated microbiome phenotypes from the RILs and NILs are mediated by effects of proanthocyanidin (condensed tannins) in the seed. Here the condensed tannin

production defect in pooled seed from tannin-negative RILs was complemented by introducing purified condensed tannins extracted from pooled tannin-producing RILs or condensed tannins extracted from hardwood trees (quebracho tannin) into fermentations of grain from the non-tannin RIL lines (Fig. 6a). The amount of added tannin extract was based on the average tannin concentration in tannin-producing RILs. Five microbiomes from the above experiments (Fig. 5) showing the most significant difference (smallest *p* value from PERMANOVA analysis) between tannin-positive RIL pool and non-tannin RILs pools (Supplementary Fig. 4) were used individually in the fermentations for molecular complementation.

Weighted UniFrac distances of β-diversity of the microbiome data after fermentation (Fig. 6b) were compared by subject across the different treatment groups, using the data from fermentation of powdered seed from the pool of tannin-positive RILs as a reference for distance. As expected, dot plots (Fig. 6b) of the distances of the microbiomes from the treatment groups show that microbiome composition from tannin-negative group (seed from non-tannin RILs only) was the most distant from the tannin-positive RIL reference whereas composition of the microbiome in fermentations of the two tannin-complemented fermentations (seed from non-tannin RIL lines supplemented with sorghum tannin extract or quebracho tannin extract) had intermediate distances between the tannin-positive and tannin-negative RILs (Kruskal–Wallis test followed by post hoc pairwise multiple comparisons using Dunn's Test). Thus, the introduction of condensed tannins extracted from sorghum or quebracho appears to diminish the microbiome phenotype of tannin-negative RILs, shifting overall microbiome composition toward that from fermentation of tannin-positive RIL lines.

Detailed analysis of the microbiome differences across complementation groups was done by comparing abundances of individual genera in the fermentation reactions from microbiomes of multiple subjects across the treatment groups (Fig. 6c). The abundance of 15 genera were different in the tannin-positive RIL pool compared with non-tannin RIL pool across 5 microbiomes ($q < 0.1$, rANOVA followed by FDR correction, Fig. 6c). Organisms such as *Lachonoclostridium, Roseburia, Alistipes, Faecalibacterium* and *Flavonifractor*, which showed significant increases in abundance with RILs carrying wild-type *Tan1* and *Tan2* loci (Fig. 5) were also more enriched in the tannin-positive RIL group in at least four out of five microbiomes (Fig. 6c). Introduction of condensed tannin extracts from tannin-producing RILs or from quebracho into fermentations of seed from non-tannin RILs also drove increases in abundances of these same taxa (Fig. 6d). The quebracho condensed tannin extracts, which are more refined than our sorghum tannin extracts, showed remarkably similar changes to the tannin-positive sorghum RILs, and correlation analysis of taxonomic abundances between fermentations of the tannin-positive RIL pool and fermentations of the tannin-negative RIL pool complemented with purified quebracho tannin was highly significant (Pearson correlation coefficient of 0.83, $p = 0.00014$). Thus, purified condensed tannins can complement the defects of seed from tannin-deficient RILs in the microbiome fermentations.

## Condensed tannins stimulate growth of *Faecalibacterium prausnitzii* in the context of a microbiome and in pure culture

While several organisms respond to both genetic variation at the *Tan1* locus and to molecular complementation by addition of condensed tannins, abundance differences in *Faecalibacterium* in response to condensed tannins were consistently observed in microbiomes from multiple human donors in both the genetic analyses and molecular complementation (Figs. 4–6). We therefore used this organism as an indicator to determine if condensed tannins can directly stimulate its growth. Using minimal fermentation media alone or minimal media

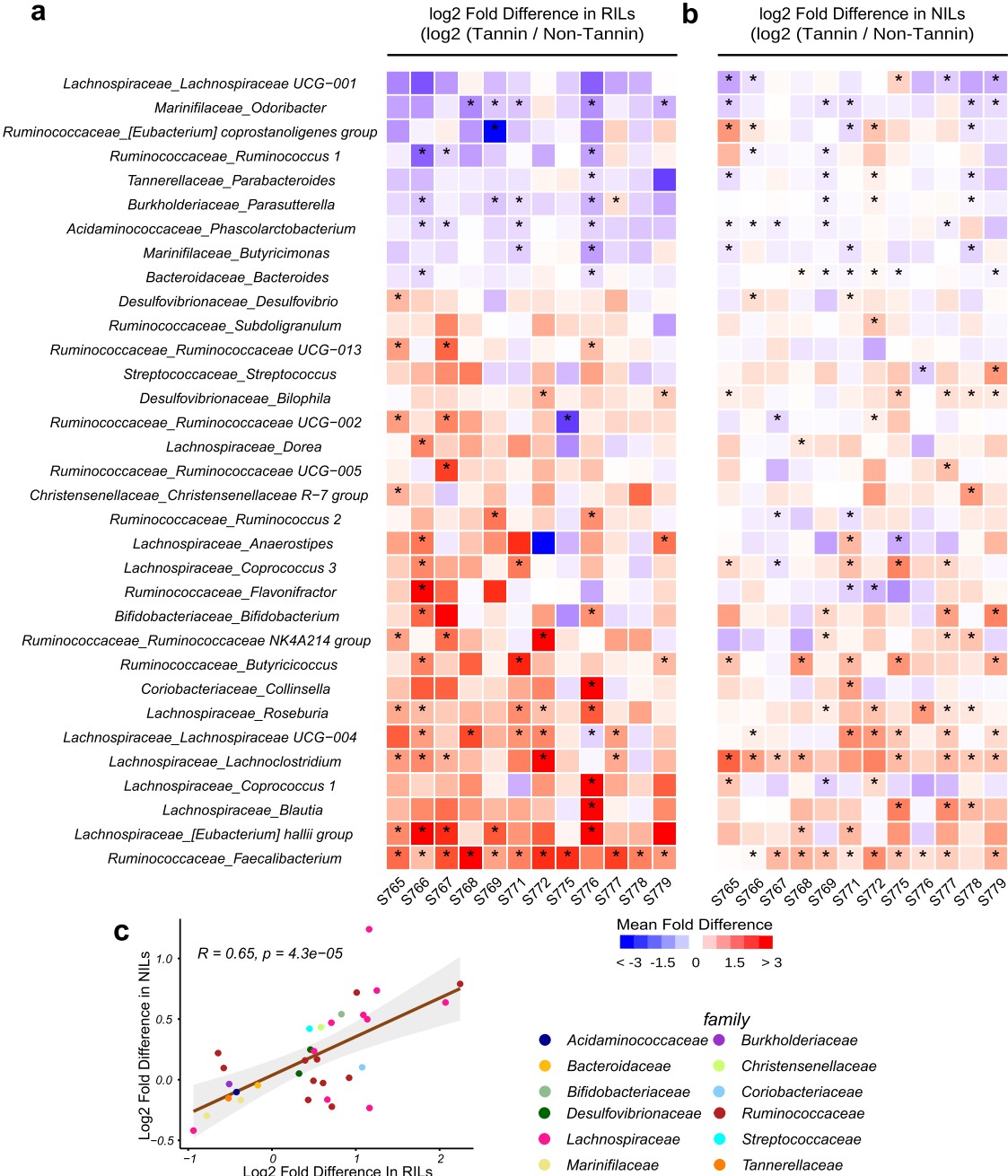

**Fig. 5 | Fermentation profile of RILs and NILs differing in tannin production across microbiomes from multiple human subjects. a** Heat map of the genera that showed significant overall effects (unadjusted $p < 0.05$; two-sided rANOVA) of abundances in AiMS fermentations of tannin-positive RILS (*Tan1/Tan2*) relative to tannin-negative RILS (*tan 1-b/Tan2*; *Tan 1/tan-2-c*; *tan-1-b/tan-2-c*) across the microbiomes of 12 different human subjects. The heat map is colored based on the log2-transformed fold difference of abundances of each taxon in tannin-positive RILs vs tannin-negative RILs and the color key is indicated at the bottom of the heat map. **b** Heat map of the same genera and subject microbiomes from panel A based on data from AiMS fermentations of Near-Isogenic Lines (NILs) created by crossing the *tan 1-b* mutation into three different tannin-positive parental lines. In this panel, the mean log2-transformed fold difference of abundances of the microbial genera in AiMS reactions of tannin-positive parental lines relative to tannin-negative NIL derivatives of each parent across each microbiome are colored according to the same color key at the bottom of the heat map. Statistical significance of differences was determined by two-sided Wilcoxon test corrected by Benjamini–Hochberg pairwise comparisons; $p < 0.05$ and those genera showing significance in one or more microbiomes are denoted by an asterisk. **c** Scatter plot of log2 fold difference of taxa in AiMS reactions of tannin-positive/tannin-negative NILs versus tannin-positive/tannin-negative RILs. The plot depicts the average log2 fold difference in each genus in data from the RILs on the X axis versus average log2 fold difference for the same genus in data from the NILs on the Y-axis. Each genus is colored according to its taxonomic family with a color key shown to the right. The Pearson correlation coefficient and two-tailed $p$ values are indicated on the dot plot. Shaded regions represent 95% confidence intervals. Source data are provided as a Source Data file.

supplemented with quebracho tannin, we measured the absolute levels of *Faecalibacterium prausnitzii* by species-specific qPCR in fermentations inoculated with human microbiomes or with pure cultures of *F. prausnitzii*. Four of the five microbiomes showed that the absolute levels of *F. prausnitzii* increased by at least 1-logarithm when condensed tannins were introduced (Fig. 7a). Similarly, addition of condensed tannins to pure cultures of *F. prauznitzii* also increased levels of the organism by >0.5 logarithm (Fig. 7a) during incubation. Notably,

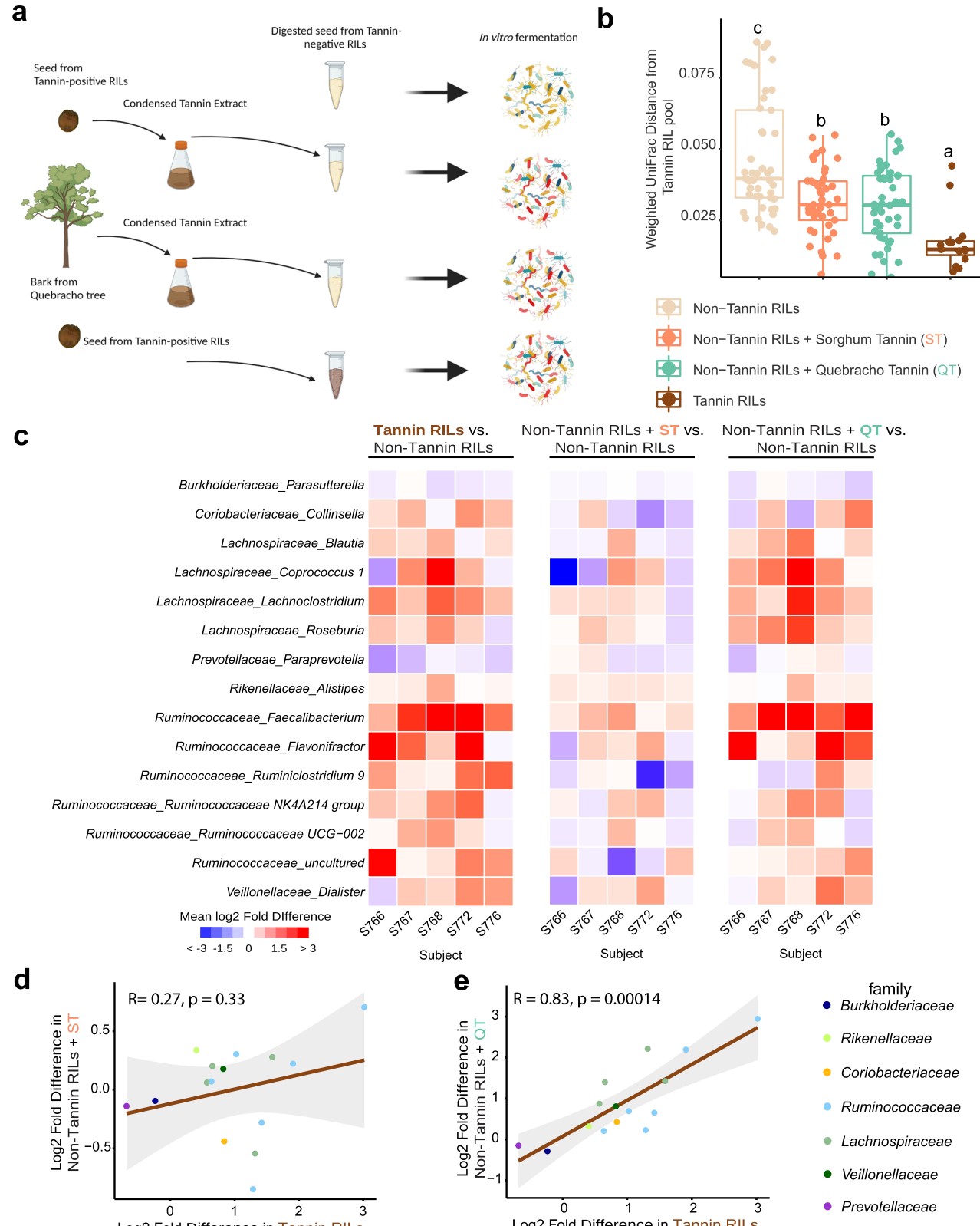

growth stimulation was higher in the context of the complex microbiomes compared to monoculture, suggesting the potential for mutualism between *F. prausnitzii* and other bacteria in utilizing tannin as a growth substrate.

Lastly, we confirmed the ability of pure cultures of *F. prausnitzii* to degrade condensed tannins by introduction of the brown-colored quebracho tannins into solid growth media. Here, fermentation media

with brown-colored quebracho tannin extracts were overlaid onto 24-h cultures of *F. prausnitzii* streaked onto LYBHI agar (Fig. 7b). After 48 h of additional incubation, zones of clearing were distinctly evident around areas of dense growth as well as individual colonies, indicating that this organism can degrade the brown-colored tannins to colorless products.

**Fig. 6 | Molecular complementation of tannins in AiMS reactions with tannin-negative RILs. a** Schematic diagram of the molecular complementation experiment. Five microbiomes showing the most significant difference between tannin-positive RIL pool and non-tannin RILs pools were used individually in the fermentations for molecular complementation. The diagram was created with BioRender.com. **b** The UniFrac distances of each fermentation samples within each microbiome are plotted from the corresponding microbiome in tannin RIL treatment samples (Tannin RILs, $n = 15$; other 3 groups, $n = 45$). The different letters (**a**–**c**) above each treatment group indicate statistical significance between treatments determined by two-sided pairwise Kruskal–Wallis test corrected by Dunn's multiple comparisons ($p < 0.05$). Displayed are the interquartile range (IQR; boxes), median (line), and 1.5 IQR (whiskers). **c** Heat map of the mean log2-transformed fold difference of genera that showed significant overall effects (unadjusted $p < 0.05$; two-sided rANOVA) of tannin-positive pool, tannin-negative pool plus sorghum tannin extract (ST) and tannin-negative pool plus quebracho tannin extract (QT)

relative to tannin-negative pool, in each microbiome respectively. **d** Taxa showing significant difference between tannin-negative control and tannin-negative RIL + sorghum tannin extract (ST) are shown. The plot depicts the average log2 fold difference in each genus in data from the positive control (tannin-containing RILs) on the *X* axis versus average log2 fold difference for the same genus in data from sorghum tannin extract treatment group on the *Y*-axis. **e** Scatter plot shows taxa with significant difference between tannin-negative control and tannin-negative RIL + quebracho tree tannin extract (QT). The plot depicts the average log2 fold difference in each genus in the data set from the positive control (tannin-containing RILs) on the *X*-axis, and data from the quebracho tree tannin extract treatment group on the *Y*-axis. Each genus is colored according to its taxonomic color with the color key on the right. Pearson correlation coefficient and two-tailed *p* values are indicated on the plot. Shaded regions represent 95% confidence intervals. Source data are provided as a Source Data file.

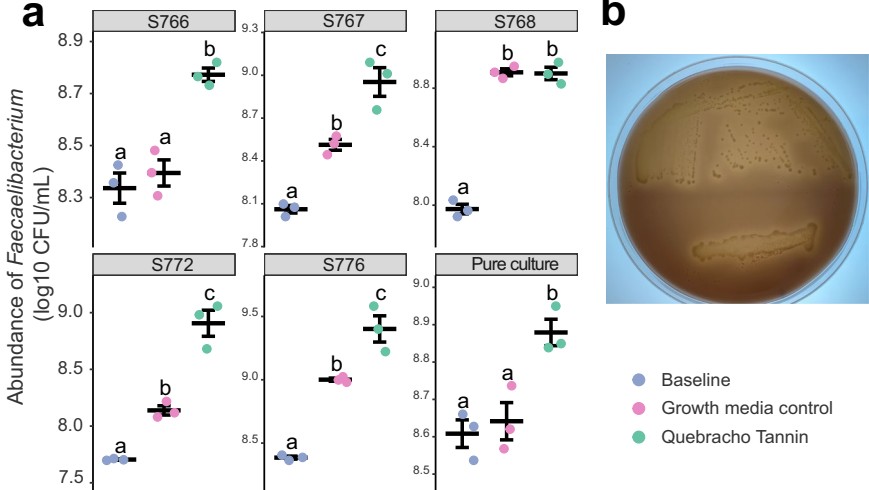

**Fig. 7 | Condensed tannin alone can stimulate growth of *Faecalibacterium prausnitzii*. a** Dot plots of absolute abundance of *Faecalibacterium prausnitzii* by qPCR in microbiome or pure culture of *Faecalibacterium prauznitzii* A2-165 with fermentation media alone or fermentation media supplemented with quebracho tree condensed tannin extract. Five microbiomes showing the most significant difference between tannin-positive RIL pool and non-tannin RILs pools were chosen, and qPCR was run on the baseline (after introduction of the stool samples) and on the samples after 16 h of incubation. The absolute level of pure culture *F. prausnitzii* A2-165 for the baseline (after inoculation with *F. prausnitzii* A2-165) and after 48 h of incubation was also determined using qPCR ($n = 3$ for each group). Data are presented as mean values ± SEM. Different letters (a–c) indicate significant

difference among different treatment groups determined by two-sided one way ANOVA corrected by Benjamini–Hochberg pairwise comparisons ($p < 0.05$).
**b** Evaluation of tannin utilization activity of *F. prausnitzii* A2-165 in agar overlay assay. LYBHI media was supplemented with 1% agar and a pure culture of *F. prausnitzii* was streaked onto the media for isolated colonies (top portion of petri dish) or in a patch (bottom portion of the plate). After 24 h of anaerobic incubation, the culture was overlaid with fermentation media supplemented with 1% agar and quebracho tree tannin. The culture was then incubated for an additional 48 h. Areas of clearing around individual colonies and the patched growth indicated degradation of the brown-colored tannin. Source data are provided as a Source Data file.

## Discussion

Our study leveraged the well-characterized population of *S. bicolor* BTx623 X IS3620C RILs to explore variation in traits that can affect the human gut microbiome. Our QTL analyses revealed a complex, polygenic genetic architecture controlling seed traits that affect the human gut microbiome fermentation patterns, with ten significant QTLs affecting different combinations of microbial taxa, concentrations of microbial fermentation products (SCFAs). Several of the QTLs corresponded to MELs with pleiotropic effects on multiple microbial taxa and their metabolites (Fig. 3), illustrating how variation in seed traits can have dramatic effects on the human gut microbiome.

Genetic diversity between the BTx623 and IS3620C parents represents only a fraction of the total genetic diversity of the *S. bicolor* species[31,32] and consequently, seed traits that affect human gut microbiome and the genetic architecture driving them in sorghum are likely to be far more diverse than the ten QTL identified in our study. Indeed, a more comprehensive catalog of these traits and the underlying genetics will emerge from genetic analysis of these traits in

populations such as the Sorghum Association Panel[33], which captures genetic diversity of sorghum from all four of the major subpopulations representing the major domestication events. The use of multiple, diverse human gut microbiomes for AiMS phenotyping will also provide a more complete catalog of traits that affect gut microbiome fermentation because the ability to detect diverse types of traits is likely dependent on species composition of the human microbiome used for AiMS phenotyping. However, an important observation from our work was that MELs affecting multiple taxa from a single microbiome tend to have significant effects across diverse microbiomes from multiple human subjects. For example, the MEL on chromosome 4 that we localized to *Tan1* affected 8 different genera from the microbiome of subject 765, but reactions with NILs (Fig. 5), RIL pools (Fig. 6), and molecular complementation experiments (Figs. 6 and 7) all showed significant effects of *Tan1* or condensed tannins themselves on microbiomes of multiple subjects. While the overall effects of condensed tannins were unique to each microbiome, a subset of microbial taxa, including *Faecalibacterium*, *Flavonifractor*, and other

members of the *Ruminococcaceae*, shared similar responses to tannins in the context of diverse microbiomes from additional human subjects (Figs. 5 and 6). Thus, while MELs can affect microbiome-specific combinations of organisms, they can also display shared effects on the same or similar organisms across diverse human microbiomes. This principle is informative because it suggests that our hierarchical approach using small numbers of human donor microbiomes to identify candidate loci affecting multiple microbes (MELs) followed by validation using larger numbers of donor microbiomes may be an efficient strategy for comprehensive analysis of seed traits that affect fermentation profile by the human gut microbiome.

Our QTL analyses, detailed genetic analyses of variation at *Tan1* and *Tan2*, and molecular complementation experiments collectively illustrate how genetic variation can translate into seed traits that affect the gut microbiome. Here, the genetic analyses with RILs and NILs localized variation affecting multiple microbiome traits in AiMS reactions to the *Tan1* and *Tan2* loci, showing that RILs carrying wild-type alleles at both the *Tan1* and *Tan2* loci increase abundances of 8 different microbial taxa while RILs carrying the null *tan1-b* allele from BTx623, the null *tan2-c* allele from IS3620C or both reduce abundances of these organisms. The null *tan1-b* and *tan2-c* alleles can individually (and collectively) block expression of the entire proanthocyanidin pathway and lead to seeds that lack condensed tannins whereas segregation of wild-type *Tan1* (inherited from the IS3620C parent) and *Tan2* (from the BTx623 parent) in the RIL progeny produce lines capable of expressing the entire proanthocyanidin pathway and that produce condensed tannins in the seed. Our molecular complementation experiments, where purified condensed tannins were introduced into AiMS reactions of seed from tannin-negative RILs, showed that condensed tannins themselves can bring about the same changes in microbiome composition. We further demonstrated that the introduction of condensed tannins into complex microbiomes or into pure cultures of *F. prauznitzii* increases the absolute levels of *F. prausnitzii*, implying that the condensed tannins stimulate growth of this organism. While microbial pathways for degradation of condensed tannins are not known, we showed that *F. prauznitzii* can degrade and grow on condensed tannins in pure culture (Fig. 7). Thus, genetic variation in genes controlling tannin synthesis affects condensed tannin content in the seed, a component that can serve as a growth substrate for specific taxa such as *F. prausnitzii* and other members of the *Ruminococcaceae* in the microbiome.

There are striking parallels between the tannin-stimulated microbes in our study and the effects of condensed tannins on microbiomes observed in several different studies. For example, introduction of quebracho tannins into in vitro fermentations containing growth media with various food components and human stool samples showed increased abundances of members of the *Lachnospiraceae*, *Ruminococcaceae* (including *Faecalibacterium*), *Christensenellaceae*, and *Peptostreptococcaceae*, and decreased the abundance of *Parabacteriodes*[34]. Similarly, feeding studies in swine and poultry show that introducing high-tannin sorghum into the feed stimulates increased abundances of members of the *Lachnospiraceae*, *Ruminococcaceae* (including *Faecalibacterium*), and *Peptostreptococcaceae* in fecal and cecal microbiota[35–37]. Work in ruminants shows that condensed tannins can reduce methane production and may do so by altering the gut microbiome although the associated organisms are yet unknown[38]. Human clinical trials also show that introducing high-tannin sorghum into the diet induces increased abundances of *Faecalibacterium prausnitzii*[39]. We have now shown that *F. prausnitzii*, can directly degrade condensed tannins (Fig. 7b) and use condensed tannins as growth substrates in monoculture (Fig. 7a). Given the depletion of this organism that is typically observed in individuals with inflammatory bowel diseases[25,26], our findings suggest that introduction of foods containing condensed tannins may help to preserve this organism in IBD patients.

Tannin production is uncommon among domesticated grains, being observed in sorghum and finger millet and it is believed that the trait was lost in other major grains during domestication due to selection against tannin-mediated phenotypes such as bitter taste[40,41]. The trait may have been maintained in sorghum, however, because tannin content contributes to agronomically-important characteristics such as resistance to bird predation[28,42]. In many regions of Africa, where sorghum is a staple in the human diet, local cultivation and consumption of tannin versus non-tannin lines correlates geographically with the intensity of bird predation and allelic variation at *Tan1* and *Tan2* are the major drivers of tannin and non-tannin phenotypes in locally cultivated sorghum varieties[28,43]. Remarkably, this study also identified geographic bias in allelic variation of human bitter taste receptor genes, where human populations cultivating high-tannin sorghum to reduce bird predation hive higher frequencies of human taste receptor alleles that decrease bitter taste perception[28]. Our work appears to further expand these complex allelochemical interactions by illuminating the effects of tannins on beneficial gut microbes, begging the question of whether disproportionate health outcomes may arise in human populations consuming high-tannin vs low-tannin sorghum due to the effects of tannins on beneficial gut microbes such as *F. prausnitzii*.

Significant QTLs with characteristics of MELs were also identified on chromosome 2, chromosome 3, and chromosome 5 (Fig. 3). We localized potential candidate genes at these MELs by a two-step process, looking for genes that (a) are expressed in sorghum seed and (b) contain allelic variation between two parental lines (Supplementary Data 7). The candidates include a putative cytochrome P450 associated with synthesis of sesquiterpenes (Sobic.002G273600), putative gibberellin 3-beta-dioxygenase (Sobic.003G045900) and abscisic acid insensitive3 (ABI3) transcription factor (Sobic.003G398200) that modulate gibberellic acid (GA) and abscisic acid (ABA) signaling in seed development, and a putative xylanase inhibitor protein precursor (Sobic.005G098700) (Fig. 3) that could influence microbial xylanases in the AiMS reactions. Gibberellin 3-beta-dioxygenase can alter grain composition, including starch content, by its effects on α-amylase[44] and the abscisic acid insensitive3 (ABI3) transcription factor also plays important roles in seed lipid and protein content as well as development[45]. Xylanase inhibitor proteins have been described in several food crops where they function to inhibit the degradation of plant cell wall arabinoxylans by invasive species of xylanase-producing bacteria and fungi[46]. In the context of our AiMS reactions, these xylanase inhibitors could affect the ability of organisms to grow on xylans from the seed pericarp.

While much work remains to validate effects of variation in these putative candidates, our study illustrates how complex trait analysis can be used to identify loci in food crops where genetic variation affects seed composition and fermentation patterns by the human gut microbiome. This approach can apply to any food crop to develop a comprehensive catalog of seed traits that affect the human gut microbiome. We believe such approaches will also pave the way for use of seed traits with major effects on beneficial gut microbes as traits for crop improvement strategies that can have profound outcomes with respect to human health.

## Methods

### Germplasm

A total of 294 F7-8 RILs derived from a *S. bicolor* BTx623 by IS3620C cross[22,47] were grown in the Greenhouse Innovation Center at the University of Nebraska-Lincoln, Lincoln, NE, USA. The greenhouse growout was planted on September 18, 2017 and harvested on February 6, 2018. The temperature was maintained between 26.6 °C and 27.8 °C during day light hours and between 21.1 and 23.3 °C during night hours with a target relative humidity of 30%. Supplemental LED lighting was employed to maintain total photosynthetically active

radiation (PAR) at or above 230 μmol m$^{-2}$ s$^{-1}$. All plants were watered to field capacity. Heads were bagged to ensure self-pollination of individual RILs. The B Wheatland sorghum line containing tannins was developed through cross-pollination of 'B Wheatland' by 'B SD106', the source of the tannin trait. The resulting F1 progeny were allowed to self-pollinate, and F2 progeny containing tannins were identified and backcrossed to the recurrent parent B Wheatland. The B Tx631 sorghum line was cross-pollinated with two different tannin-producing lines, 'Waconia' and 'B KS5'. The resulting F1 progenies were allowed to self-pollinate, and F2 progenies were backcrossed to the recurrent parent B Tx631 and B KS5, respectively. The tannin trait was identified in homozygous lines in the generation following second round of self-pollination.

### In vitro digestion

Two grams of seeds from each line were milled using a high-throughput ball mill (2025 GenoGrinder; SPEX SamplePrep, Metuchen, NJ, USA). Twenty milligrams (±0.5 mg) of each flour per replicate was dispensed into 1 mL-deep wells in 96-well plates using an automated powder dispenser (Flex PowderDose; Chemspeed Technologies AG, Füllinsdorf, Switzerland). Each flour was dispensed in triplicates. The dispensed flour was mixed with 425 μL of water for 15 min for complete dispersion and steamed for 20 min. Forty-five microliters of 0.5 M HCl + 10% (w/v) pepsin (P7000; Sigma, St. Louis, MO) were added to the samples and incubated at 37 °C for 1 h. Then 25 μL of 0.5 M sodium maleate buffer (pH = 6, containing 1 mM CaCl$_2$), 40 μL of 0.5 M NaHCO3, 40 μL of 12.5% (w/v) pancreatin (P7545; Sigma, St. Louis, MO) + 4 % (w/v) amyloglucosidase (E-AMGDF, 3260 U/mL, Megazyme) was added into samples before incubating at 37 °C for 6 h.

After digestion, samples were transferred to 96-well dialysis plates (MWCO 1,000; DispoDialyzer; Harvard Apparatus, Holliston, MA, USA) and dialyzed against 5 gallons of distilled water for 72 h at 4 °C with freshwater changes at 12-h intervals. During dialysis, each well was stirred individually with tumble stir bars in each well using a tumble stirrer (VP 710L V&P Scientific, San Diego, CA, USA). Following dialysis, the retentate was transferred into 1 mL-deep wells in 96-well plates and stored in −80 °C until fermentation.

### Fecal donor and in vitro fecal fermentation

Fresh fecal samples from 12 healthy adults (3 females and 9 males, ages 23–41) with no history of gastrointestinal abnormalities and no prebiotic, probiotic, or antibiotic consumption within the past 6 months were collected using a commode specimen collection kit (Fisher Scientific, NH, USA). All procedures involving human subjects were approved by the Institutional Review Board of the University of Nebraska−Lincoln before initiating the study (20160816311EP). Informed consent was obtained from all subjects prior to fecal collection. A 1:10 fecal slurry was prepared in an anaerobic chamber (5% H$_2$, 5% CO$_2$, and 90% N$_2$; Bactron X, Sheldon Manufacturing, Cornelius, OR, USA) within 2 h of stool collection by adding sterile 10% glycerol in phosphate-buffered saline, pH 7.0 (1:9, w/v) and mixing with a stomacher for 4 min prior to storing at −80 °C until fermentation.

In vitro batch fermentations were performed inside an anaerobic chamber. Four hundred and twenty-five microliter of retentate was mixed in a 1 mL-deep wells of a 96-well plate with fifty microliters of 10X fermentation medium containing (per liter): 1 g Bacto casitone, 1 g yeast extract, 2 g K$_2$HPO$_4$, 3.2 g NaHCO$_3$, 3.5 g NaCl, 1 mL hemin solution (KOH 0.28 g, 95% Ethanol 25 mL, hemin 100 mg and ddH$_2$O to 100 mL), 0.05 g bile salts, 0.5 g/L cysteine HCl, 0.6 mL resazurin (0.1%), 10 mL ATCC trace mineral supplement, 3.6 mL VFA solution (17 mL acetic acid, 1 mL n-valeric acid, 1 mL iso-valeric acid, 1 mL iso- butyric acid mixed with 20 mL of 10 mM NaOH), 10 mL ATCC vitamin supplement and 1 mL vitamin K-3 solution (0.14 g vitamin K-3 in 100 mL 95% ethanol)[48]. The substrate was then reduced using 25 μL of Oxyrase (Oxyrase Inc, Mansfield, OH, USA) before inoculation with 50 μL of fecal slurry. In vitro fermentations were incubated at 37 °C for 16 h. After fermentation, samples were centrifuged at 4000 × g for 10 min. Pellets and supernatants were stored at −80 °C until further processing. Fermentation media without seed substrate inoculated with fecal slurry at 0 h and incubated at 37 °C for 16 h were used as microbiome baseline controls and media-only fermentation controls, respectively.

### Validation of effect of tannin genotypes in RILs and in NILs

RILs were grouped by haplotype of markers linked to *Tan1* and *Tan2* and were randomly selected and pooled within each haplotype group (*Tan1*/*Tan2*: 41 lines (27.04 mg catechin equivalents (CE)/g), *Tan1*/*tan2-c*: 15 lines (0.36 mg CE/g), *tan1-b*/*Tan2*: 15 lines (0.25 mg CE/g), *tan1-b*/*tan2-c*: 15 lines (0.32 mg CE/g)) (Supplementary Data 6). Pooled RILs groups and NILs were digested and used as substrate for fermentation with 3 replicates across 12 human microbiomes[49].

### Tannin complementation experiment

Sorghum tannin extract (ST) (492.21 mg CE/g) from pooled sorghum RILs in haplotype of *Tan1*/*Tan2* was extracted using the method from Barros et al.[50] A commercial quebracho wood tannin extract (QT) was gifted from Silvateam Spa (San Michele di Mondoví, Italia). The extraction method for QT was natural hot water extraction. Two tannin extracts were added to the fermentations after the in vitro digestion and dialysis steps. A tannin-negative RILs pool was made by combining lines with haplotype of *Tan1*/*tan2-c*, *tan1-b*/*Tan2*, and *tan1-b*/*tan2-c* (Supplementary Data 6). Tannin-negative RILs were then complemented with ST or QT based on the tannin content in tannin-positive RILs. Tannin-negative RILs, tannin-negative RILs + ST, tannin-negative RILs + QT, and tannin-positive RILs were each inoculated with each of five human fecal microbiomes with the most significant responses between tannin-negative RILs and tannin-positive RILs sorghum observed in in vitro fermentations (smallest *p* value from PERMANOVA analysis) with three replicates.

### Tannin enrichment experiment

The commercial quebracho wood tannin extract (QT) in the 1X fermentation media was used as substrate (3.33 mg/mL) for the enrichment experiment. Fifty microliters of the same five human fecal microbiome used in tannin complementation experiment was inoculated into 500 μL fermentation media with or without QT and were incubated at 37 °C for 16 h with three replicates. Fifty microliters of the overnight culture of type strain *F. prausnitzii* A2-165 in LYBHI broth[26] was inoculated into 500 μL fermentation media with or without QT and were incubated at 37 °C for 48 h with three replicates. The overnight culture of type strain *F. prausnitzii* in LYBHI broth was also streaked on the LYBHI 1% agar plate and incubated at 37 °C for 24 h. Fermentation media plus 1% agar and QT (3.33 mg/mL) was cooled to 40 °C and then poured on top of the LYBHI plate that had *F. prausnitzii* colonies.

### DNA extraction and 16S rRNA gene sequencing

DNA was extracted from the fecal pellets using the BioSprint 96 workstation (Qiagen, Germantown, MD) and the BioSprint 96 one-for-all Vet kit with the addition of buffer ASL (Qiagen, Germantown, MD) and bead beating[51]. The V4 region of the bacterial 16S rRNA gene was amplified from each sample using the dual-indexing sequencing strategy[52].

### 16S rRNA gene sequencing processing

Paired-end sequences were analyzed using Quantitative Insights Into Microbial Ecology (QIIME) program (version 2)[53]. Sequences were truncated (220 bases for forward reads and 160 bases for reverse reads) and denoised into amplicon sequence variants (ASVs) using DADA2[54]. All ASVs were assigned with taxonomic information using pre-fitted sklearn-based taxonomy classifier SILVA database (release 132)[55,56] and were then binned at genus level and transformed to

relative abundance by dividing each value in a sample by the total reads in that sample. A neighbor-joining tree of representative sequence was generated using MUltiple Sequence Comparison by Log-Expectation (MUSCLE).

## Quantification of *Faecalibacterium prausnitzii*

The absolute abundance of *Faecalibacterium prausnitzii* was quantified using quantitative PCR method using primers (FprauF: TGAG-GAACCTGCCTCAAAGA; FprauR: GACGCGAGGCCATCTCA) described by Lindstad et al.[57]. Quantitative PCR reactions in duplicates were prepared in a 10 μL volume containing 5 μL 2× SYBR Green, 3 μL nuclease-free water, 1 μL primer mix (a mixture of forward and reverse primer of 5 μM each), and 1 μL DNA template. Thermocycling conditions included: (i) an initial denaturation step of 5 min at 95 °C; (ii) 40 cycles of 20 s at 95 °C, 25 s at annealing temperature at 63 °C, and 30 s at 72 °C; (iii) one cycle of 15 s at 95 °C; (iv) one cycle of 30 s at 60 °C; (v) one 20-min interval to generate a melting curve. The cycle threshold of each sample was then compared to a standard curve made by diluting genomic DNA from type strain *F. prausnitzii* A2-165.

## Seed color analyses

Six individual grains from each line of the RIL population were scanned using an EPSON Perfection V600 scanner. Image analysis was conducted using a set of scripts for automatic seed image analysis (https://github.com/alejandropages/SLHTP). The average red, green, blue value from each line was extracted and the first principal component of RGB value was used as trait for QTL analysis.

## Tannin content analyses

Tannin content of each line was measured using Vanillin/HCl method[58]. In short, 0.05 g of each ground sorghum flour was accurately weighed and mixed with 1% HCL in methanol and incubated at 30 °C for 20 min. After centrifugation at 2000 × g for 4 min, 0.125 mL of aliquots from the supernatant was mixed with vanillin regent (0.5% vanillin and 4% HCL in methanol) and incubated at 30 °C for 20 min. Another 0.125 mL of aliquots from the supernatant was mixed 4% HCL in methanol and incubated at 30 °C for 20 min and was used as blank. The absorbance was then measured at 500 nm against the blank. One mg/mL catechin hydrate (Sigma, St. Louis, MO) was used to generate standard curve.

## SCFA analyses

SCFA (acetate, propionate, butyrate and valerate) and branched chain fatty acids (BCFA; iso-butyrate and iso-valerate) from fermentation samples were analyzed by gas chromatography[59]. Briefly, 100 μL aliquots of fermenteation supernatant was mixed with 100 μL of 7 nM 2-Ethylbutyric acid and 100 μL of 9 M sulfuric acid and 500 μL of diethyl ether. It was then homogenized with a vortex mixer and then centrifuged at 10,000 g for 2 min. One microliter of diethyl ether layer was then analyzed using 8890 gas chromatography system (Agilent Technologies, Santa Clara, CA).

## Genetic map construction and QTL mapping

A genetic map was constructed using data from 616 informative SNP markers from the BTx623 × IS3620C RILs previously reported by Kong et al.[22] and the ASMap package version 1.0-4 in R[60]. A non-parametric interval QTL mapping procedure was employed to identify a single QTL model for each phenotypic trait using the R/qtl package version 1.47-9[61]. LOD score significance level threshholds were also calculated based on 1,000 permutations of the data with a single QTL genome scan per permutation. Significant QTL and suggestive QTL were then identified using the threshold of $p < 0.05$ and $p < 0.1$, respectively. Bayes credible interval probability was calculated using $p = 0.95$.

## Candidate genes mining

To get functional insights into MELs, genes that were highly expressed in sorghum seed and have sequence variations between two parental lines were identified. In short, an RNA-Seq data were downloaded from phytozyme v13[62] for all expressed genes in seed between the flanking genetic markers for each MEL. Tissue-specific expression of each gene was normalized using fragments per kilobase of transcript per million mapped reads (FPKM) values by dividing the average value of gene expression for that gene across all tissue types. Genes expressed in seed were further filtered to identify gene models with sequence variation in one of the two parental lines classified as having moderate or high impact on gene function using public resequencing data[62].

## Statistical analysis

All analyses were performed using R version 4.0.4 and Rstudio version 2022.07.1[63,64]. Bacterial community β-diversity in Bray-Curtis and in UniFrac distance was calculated using rarefied amplicon sequence variant (ASV) data with the phyloseq and vegan packages[65,66]. Differences in the microbiome communities were compared by PERMANOVA using the Adonis function in vegan version 2.5-7. Bacterial genera abundances and SCFA production were compared by Wilcoxon test or Kruskal–Wallis test. Data were visualized using Interactive Tree Of Life (iTOL) v5 and R packages including: ggplot2 version 3.3.3, ggpubr version 0.4.0, and Complexheatmap packages version 2.4.3[67–70]. All the figure illustrations are created by BioRender.com.

## Reporting summary

Further information on research design is available in the Nature Research Reporting Summary linked to this article.

## Data availability

The DNA sequencing reads for this study are available in the NCBI SRA database as project accession PRJNA801694. Genotype data of the RILs can be found at Figshare [https://doi.org/10.25387/g3.6304538][22]. All ASVs were assigned with taxonomic information using pre-fitted sklearn-based taxonomy classifier SILVA database [https://www.arb-silva.de/documentation/release-132/][55,56]. Source data are provided with this paper.

## Code availability

The code used in the analysis can be found at GitHub [https://github.com/qinnanyang/SorgRIL].

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

## Acknowledgements

This research was supported by funds from the Jeff and Tricia Raikes Foundation, the Bill and Melinda Gates Foundation, and the Don Dillon Foundation to A.K.B., funds from the Hogemeyer Family Foundation and the McConnell fund to J.C.S., and USDA-ARS project 3042-21220-033-00D. Q.Y. was supported in part by scholarship from the China Scholarship Council and N.K. was supported in part by Foundation for Food and Agriculture Research (FFAR) and the FFAR Fellows Program. This work was completed utilizing the Holland Computing Center of the University of Nebraska which receives support from the Nebraska Research Initiative. We thank Alejandro Pages for assistance and methods development in sorghum seed phenotyping, Dr. Keting Li and William McQueney for assistance with AiMS phenotyping, Bryce Askey for assistance in QTL mapping, and Christine Smith, Vicent Stoeger, and Troy Pabst for their assistance in plant care and harvesting. We thank Dr. Silvia Molino from Silvateam for kindly providing quebracho tannin extract for this experiment.

## Author contributions

Q.Y. designed and conducted experimental studies; A.B. supervised the work; Q.Y., M.V.H., and N.K. participated in participants recruitment; Q.Y. performed the primary analysis and interpretation data; S.S. and J.T. provided the sorghum NILs for validation experiment; J.S. assisted with interpreting the mapping results; D.R. assisted with experiment protocol; Q.Y. and A.B. drafted the manuscript; and all authors contributed to critical revisions and approved the final version.

## Competing interests

The authors declare no competing interests.
