## [Peer Review File · Nature Communications]

Genetic analysis of seed traits in *Sorghum bicolor* that affect the human gut microbiomeReviewers' Comments:

Reviewer #1:

Remarks to the Author:

Yang et al. provided a new approach for linkage analysis of MATs between sorghum grains and the human gut microbiome. They developed a novel in vitro microbiome screening (AiMS) methodology to quantify MAT phenotypes. Based on the genetic data and phenotypes in a sorghum RIL population, they detected multiple loci for various microbiomes. Specifically, the known Tan1 and Tan2 genes were shown to be associated with some microbiomes. This work is generally interesting. Further beneficial food molecules and microorganisms could be designed to improve the digestibility of the human gut.

Major comments

1. Line 292-295, the authors show the high-tannin sorghum grains induce higher abundances of Faecalibacterium, Christensenellaceae R7 group and Roseburia. What are the biological effects related to tannins of these specific microbiomes? Authors should design experiments to check the effects of those specific microbes with additional tannins.

Minor comments:

1. It would be better to add some introduction about the mentioned bioactive molecules of sorghum grains.

2. Line 174, 179 and 180, what is the evidence of the low recombination region on chromosome 2, 3 or 5?

3. Line 217-219, please redescribe gene functions of Tan1 and Tan2 since the MBW complex simultaneously regulate anthocyanin and proanthocyanin biosynthesis pathways. The variation in seed color should only be contributed by anthocyanins instead of proanthocyanins. In addition, Yellow seed1 is a major locus for seed color in diverse sorghum accessions; what is the difference between the two parental lines?

4. Line 311-316, it should provide references to support the putative gene functions. These candidates for affecting the other microbiomes may need to be verified by expression tests or sequence polymorphism detection in the parental lines.

5. Missing information of some abbreviations and in the figure legends.

Reviewer #2:

Remarks to the Author:

Manuscript "Complex Trait Analysis of Human Gut Microbiome-Active Traits in Sorghum bicolor: a new category of human health traits in food crops" is a research article focused on the characterization of possible connections among Sorghum genetics and the human microbiome configuration.

The document is overall interesting, however I have some concerns that need to be properly addressed/discussed (See below).

Major comments:

1- I think that the connections between genetics and taxa abundances are not well established and discussed. It is really difficult to me to comment such results without a direct/indirect metabolic link connecting host genetics to microbiome. In other words, the analyses, as they are, risk to be only a speculation. For all these reasons, I think the Authors need to better clarify how the

functions/phenotypes encoded by the host genes are associated to changes in microbial relative abundances by metagenomics (microbial functions connected to the different host phenotypes) and metabolomics (i.e. not only SCFA, but also other analyses for substrates/metabolites connected to the different host phenotypes). Indeed, metagenomics, with metagenomic species assembly, could help in identifying mechanisms underlying metabolic interactions within microbial consortia and explaining the change in the relative abundance, as expected in a proof of concept study.

2- Further, the authors need to quantify the genetic differences among the plant varieties. Is it possible that the microbiome changes are associated to the specific chromosome regions/genes/loci because they differentiate the plant varieties?

3- Also, is it possible that the microbiome differences derive from differences in nutritional components/substrates of the plant varieties? Please comment.

4- Did the Authors perform a control during fermentation (i.e. microbial communities in a medium without the plant substrates). I think is essential.

Minor:

- Please, better describe QTL for the readers that are not familiar with the analysis
- An acronym list could be useful for following the text

Reviewer #3:

Remarks to the Author:

This study presents a new approach for mapping plant QTLs that mediate interactions between diet components and human gut microbiome. Understanding how diet-microbiome-host interactions affect the process of energy generation, transportation, and supply for a full range of developmental, physical and mental activities that make up human life is an important topic in food science. This study attempts to explore the impact of food crop genetics on some aspects of these interactions. If significant plant genes are identified, this will help plant breeders select superior cultivars that contain favorable diet components beneficial to the human. In this sense, this study is innovative and potentially pave a new avenue at the cutting-edge of plant genetics and food science.

However, I have several significant concerns with the current form of the manuscript:

(1) The title: I don't this title. In my opinion, the title for an NC paper should be concise, informative and sharply clear. First, "Trait" appears twice, which is duplicated. Second, why have a subtitle? This makes the manuscript look like a review article.

(2) The Abstract was not clearly written. For example, in the sentence, "Several bioactive components of the human diet have major effects on composition and 25 function of the gut microbiome, but no systematic framework exists for understanding variation in microbiome-active components amid the vast amount of genotypic and phenotypic variation within a given species of food crop," it seems that "microbiome-active components" refers to diet components, but "Microbiome-Active Traits (MATs)" in the following sentence obviously indicates microbiome phenotypes. "Microbiome-active components" and MATs are not the same thing? I suggest that MATs need to be reworded, because it is quite ambiguous.

(3) In the Introduction, the authors talked about three distinct disciplines, plant breeding, gut microbiota, and food science in a way that lacks an integrative link and synthesis.

I recommend that the authors use a diet topic as a main streamline, around which the gut microbiota and plant QTL mapping are expanded into a unified framework. This will be more focused and more attractive to the readers.

(4) The most significant concern is about the ambiguity of experimental description. The Materials and

Methods section is too brief, with no detailed description of how the overall study was designed. Specifically, the following things are unclear to me:

a) If I understand correctly, you added flours (fermented and non-fermented treatments) from 417 RILs to donor stools to test fecal microbiome changes? This has two issues. First, have you tested fecal microbiomes before the flours were added? Second, you had 12 donors for each treatment. Did you pair 417 RILs with the same donor (with 12 replicates)? Actually, this is needed to exclude donor effects. Note that 417 RILs as described in the Materials and Methods become 294 RILs in the Abstract. Need to clarify here.

b) Diets affect the gut microbiome composition only when they are digested with various bioactive enzymes in the host body. When you did such an in vitro experiment, do you have any biological justification? This point needs to be clarified.

c) How did you choose donors? These donors are homogeneous in genetic background and demographical factors? No words were given about this important information.

I recommend that the authors use a diagram to demonstrate the procedure of the experiment, showing its each step from mapping population establishment to fecal microbiome data collection to isogenic line generation.

(5) QTL mapping for seed color. A discussion is needed to explain the functional relationship of seed color QTL with microbiome phenotypes. Their co-segregation is due to arbitrary covariation or functional covariation?

In summary, this manuscript contains information of some value, but its presentation, especially that about the experimental design and sampling strategies, needs to be detailed. The impact of host genetics on diet-induced microbiome change needs to be discussed. Several biological justifications, as outlined above, need to be clarified.

REVIEWER COMMENTS

Reviewer #1 (Remarks to the Author):

Yang et al. provided a new approach for linkage analysis of MATs between sorghum grains and the human gut microbiome. They developed a novel in vitro microbiome screening (AiMS) methodology to quantify MAT phenotypes. Based on the genetic data and phenotypes in a sorghum RIL population, they detected multiple loci for various microbiomes. Specifically, the known *Tan1* and *Tan2* genes were shown to be associated with some microbiomes. This work is generally interesting. Further beneficial food molecules and microorganisms could be designed to improve the digestibility of the human gut.

We are grateful for the positive feedback from Reviewer 1 and their appreciation of the novelty and potential applications of this study.

Major comments

1. Line 292-295, the authors show the high-tannin sorghum grains induce higher abundances of *Faecalibacterium*, Christensenellaceae R7 group and *Roseburia*. What are the biological effects related to tannins of these specific microbiomes? Authors should design experiments to check the effects of those specific microbes with additional tannins.

Thanks for the advice. We have conducted additional experiments using molecular complementation which demonstrate that addition of condensed tannin extracts to tannin-negative RILs increases abundances of many of the responsive microbes (e.g. those taxa associated with the *Tan1* peak on chromosome 4) (Figure 6). We further use qPCR specific for *Faecalibacterium* to demonstrate (Figure 7) that addition of condensed tannins in fermentation media increases the absolute levels of the *Faecalibacterium prauznitzii*. And pure cultures of *Faecalibacterium prauznitzii* show growth upon addition of condensed tannins to minimal media, and that pure cultures of *Faecalibacterium prauznitzii* can degrade condensed tannins on solid media (Figure 7). (Line 333 to Line 401).

Minor comments:

1. It would be better to add some introduction about the mentioned bioactive molecules of sorghum grains.

Thanks for the advice. We have added some additional material to the introduction regarding previously known bioactive molecules found in the grain of sorghum (Lines 69 to Line 74).

2. Line 174, 179 and 180, what is the evidence of the low recombination region on chromosome 2, 3 or 5?

In this revised version we have removed the reference to low recombination rates on chromosomes 2, 3, and 5.

3. Line 217-219, please redescribe gene functions of *Tan1* and *Tan2* since the MBW complex simultaneously regulate anthocyanin and proanthocyanin biosynthesis pathways. The variation in seed color should only be contributed by anthocyanins instead of proanthocyanins. In addition, *Yellow seed1* is a major locus for seed color in diverse sorghum accessions; what is the difference between the two parental lines?

Thank you for your insight and pointing out the confusing wording with respect to MBW regulating anthocyanin and proanthocyanidin biosynthesis pathways. We have reworded the section to describe the known roles of *Tan1* and *Tan2* in regulating the function of both anthocyanin and proanthocyanidin. We have also included some information about the roles of other loci (e.g. Y, R, and Z) and the conditions in which brown seed color can occur in lines with dominant *Tan1/Tan2* in backgrounds that permit spread of the tannins from the testa layer into the pericarp. (Line 218 to Line 229)

Neither the parental lines nor the RILs used in this study exhibited phenotypic variation (yellow seeds) consistent with segregation of alleles at the *yellow seed1* (Y) locus. Our QTL mapping analysis for seed color did not identify any signal at the *ys1* locus (Figure 4A) or other loci known to control seed color (e.g., R and Z loci). The same was true for QTL analysis of tannin content and microbiome traits that map to the *Tan1* and *Tan2* loci. In the revised version we include a brief discussion and reference to *yellow seed1* and other loci such as R and Z as additional sorghum locus known to regulate seed color in different sorghum populations (lines 239-245).

4. Line 311-316, it should provide references to support the putative gene functions. These candidates for affecting the other microbiomes may need to be verified by expression tests or sequence polymorphism detection in the parental lines.

In this revised version of the manuscript, we now provide information on sequence polymorphisms between the parental lines for candidate genes (supplemented dataset 6) and their expression patterns in seed (Figure 3) that are available in Phytozome V 13. Because the candidate genes for MELs on chromosomes 2, 3, and 5 remain speculative, we moved this section to the Discussion and use it to illustrate how genetic analysis of seed traits affecting the gut microbiome can identify a range of genetic variation (lines 449-523).

5. Missing information of some abbreviations and in the figure legends.

Thank you for pointing this out. We have added the information of abbreviations in Line 108, Line 118, Line 126, Line 171, Line 306 and in the figure legends.

Reviewer #2 (Remarks to the Author):

Manuscript “Complex Trait Analysis of Human Gut Microbiome-Active Traits in *Sorghum bicolor*: a new category of human health traits in food crops” is a research article focused on the characterization of possible connections among *Sorghum* genetics and the human microbiome configuration.

The document is overall interesting, however I have some concerns that need to be properly addressed/discussed (See below).

Major comments:

1- I think that the connections between genetics and taxa abundances are not well established and discussed. It is really difficult to me to comment such results without a direct/indirect metabolic link connecting host genetics to microbiome. In other words, the analyses, as they are, risk to be only a speculation. For all these reasons, I think the Authors need to better clarify how the functions/phenotypes encoded by the host genes are associated to changes in microbial relative abundances by metagenomics (microbial functions connected to the different host phenotypes) and metabolomics (i.e. not only SCFA, but also other analyses for substrates/metabolites connected to the different host phenotypes). Indeed, metagenomics, with metagenomic species assembly, could help in identifying mechanisms underlying metabolic interactions within microbial consortia and explaining the change in the relative abundance, as expected in a proof of concept study.

Thank you for pointing out the challenges in communicating this complex approach. The genetic associations point toward significant, causal effects of genetic variation at QTL peaks on the abundances of specific taxa in the AiMS fermentations. There are no known bacterial pathways for degradation of condensed tannins. Consequently, we developed an experimental approach (Molecular Complementation) where purified condensed tannins from sorghum or from bark of quebracho trees are added to AiMS reactions of tannin-negative RILs prior to fermentation (Figure 6). These experiments clearly demonstrate the ability of purified condensed tannins to elicit very similar microbial responses that are observed in AiMS reactions with tannin-positive RILs. We further demonstrated that addition purified tannins to AiMS reactions of microbiomes in minimal fermentation media leads to increased absolute levels of *Faecalibacterium* by qPCR (Figure 7), that addition of purified condensed tannins to pure cultures of *Faecalibacterium prauznitzii* also stimulate growth of the organism (Figure 7), and that addition of purified tannins into solid media with pure cultures of *Faecalibacterium prauznitzii* show this organism can degrade the colored tannins to colorless products (Figure 7). Finally, we added text describing how genetic variation in tannin regulators (*Tan1* and *Tan2* loci) influence tannin production in the seed, the color of the seeds, and how the condensed tannins themselves serve as a microbial growth substrate directly affecting abundances of specific taxa in the AiMS reactions. (Line 444 to Line 466)

2- Further, the authors need to quantify the genetic differences among the plant varieties. Is it possible that the microbiome changes are associated to the specific chromosome regions/genes/loci because they differentiate the plant varieties?

If we interpret your question correctly, indeed genetic variation that is present in the parental lines (plant varieties) is what drives the seed and microbial phenotypes. The RIL population was derived from progeny of two original parental lines (BTx623 and IS3620C) and we initially characterize the microbiome differences that can be detected in seed from these two parental lines (Figure 2 and Figure S2 in revised manuscript). The QTL analysis in the resulting RIL population (where each RIL carries essentially random combinations of genomic segments from each parent) allows us to associate genetic variation at specific genomic locations with differences in the microbiome profiles in AiMS fermentations. Some phenotypes, such as brown seed color, tannin production, and the microbial taxa

associated with QTL peaks on chromosomes 2 and 4, are due to what is known as transgressive segregation, where combinations of alleles from each parent are inherited at these two loci in the RILs and collectively the combination produces phenotypes (brown seed color, tannin production, and microbiome phenotypes) that are not observed in either parent. This is because recessive alleles of *Tan1* or *Tan2* completely block tannin synthesis, but when the progeny RILs inherit a dominant allele of *Tan1* from the IS3620 parent and a dominant allele of *Tan2* from the BTx623 parent, the anthocyanin/proanthocyanidin pathway is now expressed, tannins are produced, seeds are colored brown, and microbiome phenotypes result.

To further illustrate how the phenotypes segregate exclusively with variation at *Tan2* (Chromosome 2) and *Tan1* (Chromosome 4), we developed a PCA plot of each RIL based on total genetic diversity across all markers used for QTL mapping (Supplemental Figure 5). When the individual data points from each RIL in the plot are color-coded by tannin haplotype, it is clear that RILs with tannin-positive haplotypes (dominant alleles at *Tan1* and *Tan2*) are randomly distributed based on total genetic diversity of the population. Thus, the phenotypic variation (microbiome changes, seed color, and tannin content) that is specifically associated with RILs carrying dominant *Tan1* and *Tan2* haplotypes (Figure 4) occurs within a broad range of genetic diversity from either parent at other genomic loci. Consequently, genetic variation at *Tan1* and *Tan2* loci on chromosome 4 and chromosome 2 are the primary drivers of seed color, tannin content, and abundance changes in microbes (Figure 4). In fact, we did not detect significant association of brown seed color, tannin production, or increased abundances of taxa such as *Faecalibacterium* with variation at any other loci in the genome (Figure 4).

3- Also, is it possible that the microbiome differences derive from differences in nutritional components/substrates of the plant varieties? Please comment.

Yes. Our QTL analyses show that variation at multiple loci in the sorghum genome can contribute to changes in microbiome profiles in AiMS reactions (Figure 3). Indeed, variation at other sorghum genomic loci (e.g. MELs on chromosome 3 and chromosome 5) causes unique patterns of microbial variation in AiMS fermentations that are different from the microbial phenotypes associated with the MELs at *Tan2* (chromosome 2) and *Tan1* (chromosome 4) loci. Because the overall message of the manuscript emphasizes the novelty of the approach (genetic analysis in food crops can be used to identify variation in seed traits that affects the human gut microbiome), we focused our validation studies on mechanistic characterization of the MELs associated with tannin production as these loci are relatively well-characterized in sorghum with respect to seed color. As described above, we have introduced additional molecular complementation experiments to show mechanistically how variation in condensed tannin production arising from segregation of alleles at *Tan1* and *Tan2* drive microbiome changes consistent with the corresponding QTLs on these chromosomes (Figures 6 and Line 333 to Line 377 in the revised manuscript). We moved the description of candidate genes at the MELs on chromosomes 2, 3, and 5 to the discussion due to the speculative nature of the supporting data, and we use this section to simply emphasize that a broad array of genetic variation in the plant genome/seed characteristics can contribute to variation in fermentation by the human gut microbiome.

4- Did the Authors perform a control during fermentation (i.e. microbial communities in a medium without the plant substrates). I think is essential.

Yes. We have multiple controls that are built into the experimental design with the AiMS fermentation, including a control that accounts for baseline microbiome composition of the reactions before fermentation and controls that account for microbiome composition after AiMS fermentations with media only (e.g. no substrate from sorghum seed added). We use these controls primarily to ensure that changes in the microbiome occur from addition of seed components and such changes are distinct from those occurring from the media alone. These controls are now highlighted in the revised Materials and Methods section (Lines 583-585).

Minor:

- Please, better describe QTL for the readers that are not familiar with the analysis

Thank you. We have now added a graphical diagram of the overall study design along with a high-level description of how QTL analysis is used to as a robust statistical approach for associating phenotypic variation with genetic variation at specific loci in the sorghum genome (Figure 1 and Line 84 to Line 101).

- An acronym list could be useful for following the text

Thank you for the advice. We have added an acronym list in the main text. Line 697 to Line 704.

Reviewer #3 (Remarks to the Author):

This study presents a new approach for mapping plant QTLs that mediate interactions between diet components and human gut microbiome. Understanding how diet-microbiome-host interactions affect the process of energy generation, transportation, and supply for a full range of developmental, physical and mental activities that make up human life is an important topic in food science. This study attempts to explore the impact of food crop genetics on some aspects of these interactions. If significant plant genes are identified, this will help plant breeders select superior cultivars that contain favorable diet components beneficial to the human. In this sense, this study is innovative and potentially pave a new avenue at the cutting-edge of plant genetics and food science.

However, I have several significant concerns with the current form of the manuscript:

(1) The title: I don't this title. In my opinion, the title for an NC paper should be concise, informative and sharply clear. First, "Trait" appears twice, which is duplicated. Second, why have a subtitle? This makes the manuscript look like a review article.

Thank you for this great insight. We have now greatly simplified the title and removed "Microbiome-Active Traits" from the manuscript to make the text less confusing.

(2) The Abstract was not clearly written. For example, in the sentence, "Several bioactive components of the human diet have major effects on composition and 25 function of the gut microbiome, but no systematic framework exists for understanding variation in microbiome-active components amid the vast amount of genotypic and phenotypic variation within a given species of food crop," it seems that

“microbiome-active components” refers to diet components, but “Microbiome-Active Traits (MATs)” in the following sentence obviously indicates microbiome phenotypes. “Microbiome-active components” and MATs are not the same thing? I suggest that MATs need to be reworded, because it is quite ambiguous.

We agree with the reviewer and have accordingly revised the abstract and introduction. We have also removed the term “MATs” from the manuscript to reduce ambiguity. We believe the manuscript is now much more focused and emphasizes the novelty of the approach while avoiding confusion from introduction of new terms. We also included a conceptual figure (Figure 1) that illustrates the overall approach to mapping traits in sorghum that influence the human gut microbiome.

(3) In the Introduction, the authors talked about three distinct disciplines, plant breeding, gut microbiota, and food science in a way that lacks an integrative link and synthesis. conceptual figure

I recommend that the authors use a diet topic as a main streamline, around which the gut microbiota and plant QTL mapping are expanded into a unified framework. This will be more focused and more attractive to the readers.

Thank you for pointing this out. We agree and have now streamlined and focused the introduction to provide a more integrative view of the transdisciplinary approach that we are using. We also integrate this with a new section at the beginning of the results and a new conceptual figure (Figure 1) that graphically depicts the overall experimental approach.

(4) The most significant concern is about the ambiguity of experimental description. The Materials and Methods section is too brief, with no detailed description of how the overall study was designed. Specifically, the following things are unclear to me:

a) If I understand correctly, you added flours (fermented and non-fermented treatments) from 417 RILs to donor stools to test fecal microbiome changes? This has two issues. First, have you tested fecal microbiomes before the flours were added? Second, you had 12 donors for each treatment. Did you pair 417 RILs with the same donor (with 12 replicates)? Actually, this is needed to exclude donor effects. Note that 417 RILs as described in the Materials and Methods become 294 RILs in the Abstract. Need to clarify here.

We apologize for the ambiguity in the previous version of our manuscript, in particular with the inconsistent usage of 417 RILs (the total number that exist for the IS3620C x BTx623 population) and 294 RILs (the subset of the population used in this study, based on the availability of both seed and genetic marker data). We have removed all references to 417 RILs in this revised version to reduce the risk of confusion.

We do indeed include no flour added controls as part of our analysis and we have revised the manuscript to describe this control within the results section (lines 583-585).

We do indeed employ the same donor to evaluate all RILs as part of this study. This has been clarified on lines 128-133.

b) Diets affect the gut microbiome composition only when they are digested with various bioactive enzymes in the host body. When you did such an in vitro experiment, do you have any biological justification? This point needs to be clarified.

We agree with the reviewer's point here. We have revised our results section with the graphical overview and description (lines 84-101) to include the fact that our AiMS reactions include use of sequential acid hydrolysis and hydrolysis by bioactive enzymes (amylase and pancreatic enzymes) and dialysis to preprocess the whole-grain samples in a manner that is analogous to acid-mediated digestion in the stomach (acid hydrolysis), enzymatic hydrolysis in the small intestine (amylase and pancreatin) and absorption of small molecules in the small intestine (dialysis). The seed components remaining after these steps in the AiMS reactions reflect the remaining components that would enter the colon for further digestion by the massive enzymatic capacity of the human gut microbiome.

c) How did you choose donors? These donors are homogeneous in genetic background and demographical factors? No words were given about this important information. describe the samples collection

We now include additional text in methods (lines 561-566) describing the selection of donors and the protocol used for microbiome collection.

I recommend that the authors use a diagram to demonstrate the procedure of the experiment, showing its each step from mapping population establishment to fecal microbiome data collection to isogenic line generation.

Thank you for this excellent suggestion. We now include a conceptual figure demonstrating the experimental procedure as Figure 1.

(5) QTL mapping for seed color. A discussion is needed to explain the functional relationship of seed color QTL with microbiome phenotypes. Their co-segregation is due to arbitrary covariation or functional covariation?

As our new molecular complementation experiments demonstrate (Figures 6 and 7), the co-segregation is necessary and sufficient for the microbiome phenotypes. As the RILs were generated randomly from F2 population, we believe the covariation of Tan 1, Tan 2, color, tannin production, and microbiome traits is likely arbitrary and not due to any type of environmental selection or filtering. We have put a description of how covariation of haplotypes at *Tan1* and *Tan2* manifests into seed color, tannin content, and microbiome phenotypes (Line 444 to Line 466 in revised version)

In summary, this manuscript contains information of some value, but its presentation, especially that about the experimental design and sampling strategies, needs to be detailed. The impact of host

genetics on diet-induced microbiome change needs to be discussed. Several biological justifications, as outlined above, need to be clarified.

We have greatly changed the presentation, added more detail about experimental design and sampling strategies, and we have provided additional experiments that demonstrate how the genetic variation in grain (e.g. *Tan1* and *Tan2*) translates to differences in seed composition (condensed tannins) and shown that condensed tannins themselves can be used as substrates for growth of gut microbes, both in the context of a microbiome and even in pure culture. We do mention that use of additional microbiomes for mapping will increase the number of genetic associations. While examples of the effects of host genetics on diet-induced microbiome changes are indeed fascinating (e.g. variation at the LCT locus, abundance of Bifidobacterium, and milk consumption) there are only a few instances of such interactions and they are not well understood. Consequently, we feel that discussion of such effects is beyond the scope of this manuscript.

Reviewers' Comments:

Reviewer #1:

Remarks to the Author:

This revised version is much better than the previous version. The authors have provided additional information and text rearrangements, including the addition of new data: 1) Demonstration of condensed tannin extracts increase microbe abundances (Figure 6 and Figure 7). 2) Description of detailed experimental designs (Figure 1). Overall, this version addressed most of my concerns and might be acceptable.

One suggestion is that, the mostly mentioned *F. prauznitzii* can use tannins as growth substrates and degrade it. Whether is it probiotic? The mass-consuming wine also contains tannins, which are benefit for human health. Does that mean tannins change the gut microbiome and then affecting human health in a certain way? It might be discussed.

Additional it is known that Sorghum seed tannins, as a known molecular component, can reduce the farting of cattle and sheep by inhibiting the intestinal digestion ability and potential microbiome. It is helpful for slowing down the global greenhouse effect due to a large amount of methane gas in the farm. The research reveals the interactions between diets from host crops and human gut microbiome. Since authors identified Tan1, Tan2 genes and other unknown loci and found tannins could simulate *Faecalibacterium prauznitzii* growth. Discussion about this point is also welcomed.

Figure 6 legend, remove the additional "introduced".

Figure 7 legend, absence of introduction of S766-S776.

Reviewer #2:

Remarks to the Author:

Thanks to the Authors for the many clarifications. I endorsed the publication.

Reviewer #3:

Remarks to the Author:

The authors have done a good job to address my concern.

REVIEWER COMMENTS

Reviewer #1 (Remarks to the Author):

This revised version is much better than the previous version. The authors have provided additional information and text rearrangements, including the addition of new data: 1) Demonstration of condensed tannin extracts increase microbe abundances (Figure 6 and Figure 7). 2) Description of detailed experimental designs (Figure 1). Overall, this version addressed most of my concerns and might be acceptable.

We sincerely appreciate you for your time in reviewing our manuscript and offering your valuable comments.

One suggestion is that, the mostly mentioned *F. prauznitzii* can use tannins as growth substrates and degrade it. Whether is it probiotic? The mass-consuming wine also contains tannins, which are benefit for human health. Does that mean tannins change the gut microbiome and then affecting human health in a certain way? It might be discussed.

Thank you for your suggestion. As we discussed in the manuscript Line149-151, *Faecalibacterium* are increasingly being recognized as beneficial organisms in the microbiome that reduce susceptibility to inflammatory diseases. This may qualify *Faecalibacterium* as a probiotic, although there are no commercially available probiotics on the market that are based on *Faecalibacterium*, largely because the organism is difficult to grow.

We have not added any discussion about tannins in wine. Tannins in wine are hydrolyzable tannins, different in structure from the non-hydrolyzable, condensed tannins found in sorghum. Because of the structural differences, we expect that the interaction between sorghum tannins and the microbiome is very different from the interaction between wine tannins and the microbiome. Thus, we considered a discussion on tannins in wine to be outside the scope of this study.

Additional it is known that Sorghum seed tannins, as a known molecular component, can reduce the farting of cattle and sheep by inhibiting the intestinal digestion ability and potential microbiome. It is helpful for slowing down the global greenhouse effect due to a large amount of methane gas in the farm. The research reveals the interactions between diets from host crops and human gut microbiome. Since authors identified Tan1, Tan2 genes and other unknown loci and found tannins could simulate *Faecalibacterium prauznitzii* growth. Discussion about this point is also welcomed.

Thank you for your insight on the effect of condensed tannins on slowing down the global greenhouse effect. While our study is focused on how genetics of grain crops could influence the human gut microbiome, we did include a sentence (Lines 470-471) indicating that condensed tannins can

influence ruminant methane production. However, the mechanism and associated microbes remain unknown.

Figure 6 legend, remove the additional “introduced”.

We have removed the additional “introduced”.

Figure 7 legend, absence of introduction of S766-S776.

We have added additional introduction of 5 microbiomes that were chosen.

Reviewer #2 (Remarks to the Author):

Thanks to the Authors for the many clarifications. I endorsed the publication.

Thank you for your very helpful suggestions

Reviewer #3 (Remarks to the Author):

The authors have done a good job to address my concern.

Thank you for your very helpful suggestions